

# An Impact Chain-based exploration of multi-hazard vulnerability dynamics. The multi-hazard of floods and the COVID-19 pandemic in Romania

Andra-Cosmina Albulescu[1, 2], Iuliana Armaș[1]

[1] Center for Risk Studies, Spatial Modelling, Terrestrial and Coastal System Dynamics, Faculty of Geography, University of Bucharest, Bucharest, 030018, Romania

[2] Faculty of Geography and Geology, "Alexandru Ioan Cuza" University of Iasi, Iasi, 700506, Romania

*Correspondence to*: Andra-Cosmina Albulescu (cosminaalbulescu@yahoo.com, Cosmina.albulescu@uaic.ro)

**Abstract.** In the present multi-hazard-prone times, the dynamics of vulnerability across time, space, and different hazards emerges as an intriguing but challenging research topic. Within multi-hazards, both the impacts of hazards and the mitigation strategies can augment vulnerabilities, adding layers to the complexity of multi-risk assessments. Delving into these intricacies, this study aims to i) explore the multi-hazard impacts of the co-occurrent powerful river flood events and the COVID-19 pandemic in Romania, taking as reference 2020 and 2021, and ii) to analyse the trajectories in rising vulnerability that result from impacts and adaptation options, as well as their implications. The proposed framework relies on an Impact Chain that was enhanced to include new elements (i.e., augmented vulnerabilities and derived impacts) and links (i.e., connections that describe the augmentation of vulnerability); which were also used to rank the vulnerabilities based on their augmentation. The Impact Chain draws on various data and information sources, including scientific literature, the feedback of first responders, reports, legislative documents, official press releases, and news reports. This research work makes a significant contribution to the field of DRR by broadening the purpose of the Impact Chain, transforming it into a first-hand, semi-qualitative tool for analysing vulnerability dynamics.

## 1 Introduction

The third decade of the 21st century debuted with a pivotal epidemiological hazardous event that taught human communities worldwide formative and often cruel lessons. The COVID-19 pandemic acted as a powerful driver of societal change, making scientists and practitioners reconsider their approaches to health care management (Begun and Jiang 2020, Rawaf et al. 2020, Matenge et al. 2022), health systems resilience (Chua et al. 2020, Hariri-Ardebili 2020, Traverson et al. 2020, Haldane and Morgan 2021, Haldane et al. 2021), or resilience in general (Hariri-Ardebili et al. 2022), as well as multi-hazard risk management (Quigley et al. 2020, Potutan and Arakida 2020, Ali Maher 2021, Ashraf 2021, Kruczkiewicz et al. 2021, Simonovic et al. 2021, UNU-EHS 2021, Mavroulis et al. 2022, Terzi et al. 2022), and giving way to new economic challenges (Buheji et al. 2020, Kaye et al. 2020, Asare and Barfi 2021, Sikder et al. 2020, Younas and Kassim 2022).





In the field of Disaster Risk Reduction (DRR), the co-occurrence of natural hazards of various types and magnitudes amid the COVID-19 pandemic has caused a paradigm shift. Multi-hazard analysis switched its focus from analysing all the hazards that can affect an area in a given period of time, which is often called multilayer single hazard analysis (Gill and Malamud 2014) or "all-hazards-at-place approach" (Hewitt and Burton 1971), to analysing the interactions between the hazards that overlap in time and space (De Angeli et al. 2022). This shift was supported by the Sendai Framework for Disaster Risk Reduction 2015-

2030 and the Paris Agreement. A first positive outcome was the consolidation of on-point definitions of terms that were previously more flexible in their approach, as shown by the comprehensive literature review performed by Ciurean et al. (2018): multi-hazard risk and multi-risk (Zschau 2017, Gill et al. 2022).

    In the new multi-hazard-prone era, vulnerability represents a key component of multi-risk analysis due to the fact that its spatial and temporal dynamics is reshaped by the impact of multiple hazards and also by adaptive strategies. This raises significant

challenges for risk management while reinforcing vulnerability's role in portraying disasters as human constructs (de Ruiter and van Loon 2022). This study delves deeper into the changes in vulnerability under hazard-generated impacts, taking as a case study two co-occurrent, independent hazards (i.e., floods and the COVID-19 pandemic) that severely affected a European country. At the outset, it is necessary to clarify the role of impacts resulting from multiple hazards in shaping vulnerability, with illustrative recent examples from the literature. These instances bring to light a notable research gap that requires

investigation, as detailed in the following.

    Hazards generate various impacts on exposed elements with certain vulnerability levels. A particular impact has the potential to alter the vulnerability conditions that underlie another impact, whether it is caused by the same hazard or a different one. Another way to frame this issue is that the impact of a hazard changes vulnerability conditions before the recovery process reaches its end, with significant implications for the manifestation of a different hazard (de Ruiter et al. 2020). This is also

mentioned by Mohammadi et al. (2023) in relation to the functionality of a system: "Additionally, events of any size, no matter how severe, that occur after a destructive event may result in the system's functionality being reduced because the system will be more vulnerable than it was prior to the big event, due to the damages that have been imposed by the first big event." The stated situation corresponds to the third type of dynamic vulnerability identified by de Ruiter and van Loon (2022), namely, the changes in vulnerability during compounding disasters that are caused by a chain of events.

A particular situation is the one where the adaptation options or the structural measures implemented to reduce the risk associated with one hazard (de Ruiter et al. 2020) or the vulnerability to one hazard (Ward et al. 2020) have unwanted effects, increasing the risk associated with a second hazard, respectively the vulnerability to another hazard, leading to asynergies (de Ruiter et al. 2020). This means that multi-risk analyses become even more convoluted and that they have to account for interactions that act as both causes and effects; which is a tall order for both researchers and decision-makers (Reichstein et al.

2021), but it is also essential to consider in the recovery phase of the Disaster Risk Management (DRM) cycle (Mohammadi et al. 2023).

    The scientific literature provides several examples (Table 1) that point out failures of hazard management, which stem from the fact that standard operational procedures were not adapted to pandemic conditions, or from the fact that the efforts of tilting





the SARS-CoV-2 infection curve were not adapted to fit hazard management practices. In recent years, this conundrum has

become a hot topic in the field of DRM, being debated by numerous scientists (Frausto-Martínez et al. 2020, Quigley et al. 2020, Potutan and Arakida 2021, Albulescu et al. 2022, Hariri-Ardebili et al. 2022). A counterexample is given by Mavroulis et al. (2022), who present pandemic-adapted practices of emergency response focusing on the cases of the earthquakes that hit different regions of Greece in 2020 and 2021.

**Table 1. Examples of multi-hazard contexts where management was hindered by a lack of adaptation to pandemic conditions**

| Multi-hazards with location and time | References |
|---|---|
| The COVID-19 pandemic, landslides, and floods, coupled with the social problems that affected Cox's Bazar refugee camp in Bangladesh in the summer of 2021 | Patwary and Rodriguez-Morales (2021) |
| The "triple threat" of the pandemic, floods, and locusts in East Africa in the spring of 2020 | Kassegn and Endris (2021) |
| Different cyclones (e.g., Harold, Yasa, Ana, and Rolly) in the Philippines amid pandemic waves in 2020 and 2021 | Mangubhai et al. (2021), Izumi and Shaw (2022) |
| The co-occurrence of the pandemic and cyclone Amphan (May 2020) in West Bengal and in India | Majumdar and Dasgupta (2020), Pramanik et al. (2021) |
| The co-occurrence of the pandemic and Cyclone Harold (April 2020) in Vanuatu | UNDRR (2020) |
| The pandemic and droughts in the Western USA, Southeastern Australia, and Asia | Mishra et al. (2021) |
| The tornadoes that hit the Southeastern USA in April 2020, towards the end of the first pandemic wave | Andrews (2020) |


This collection of negative (Table 1) and positive examples motivates the need for an in-depth understanding of the interplay of different hazards and of the spatial-temporal changes in exposure, vulnerability, and adaptation settings. It is only by gaining a profound understanding, that we can develop new DRM models that account for pandemic conditions and acknowledge that all systems have limited and variable capacity (Terzi et al. 2022), followed by improved multi-risk management (Potutan and

Arakida 2020, UNDRR 2020, Ashraf 2021, Ishiwatari et al. 2020).

Up to date, scientific works on the interactions between natural hazards and the COVID-19 pandemic have primarily revolved around factual observations, overlooking the effects on the dynamics of vulnerability. All the examples listed above pertain to hydro-climatic hazardous events amid the pandemic, offering only factual documentation on their interactions. Narrowing down to the flood hazard, the compounded impacts of flood events and the pandemic are largely unknown and have been

described only tangentially or in short (Simonovic et al. 2020, Patwary and Rodriguez-Morales 2021, Pramanik et al. 2021, Turay 2022), although the pandemic can augment typical health-related flood impacts (e.g., injuries, gastric problems stemming from water contamination, increased stress and/or anxiety) (Simonovic et al. 2020). Instead, more literature is available on the potential effects of flood events on the dynamics of COVID-19 cases (Frausto-Martínez et al. 2020, Mavroulis et al. 2021a, b, Albulescu 2023). What is more, the augmentation or attenuation of vulnerability conditions by previous hazard

impacts (be they floods, pandemics, or other hazards) was not considered in any case study and has only been documented related to long-term processes (de Ruiter and van Loon 2022).



This study aims to address the research gap regarding the dynamics of vulnerability in a multi-hazard context by i) exploring the multi-hazard impacts of the co-occurrent extreme river flood events and the COVID-19 pandemic in Romania, taking as references 2020 and 2021, and ii) analysing the increases in vulnerability that stem from hazard impacts and adaptation options, as well as their implications. The proposed methodological framework relies on two Impact Chains: the first one is used to document the two-year unfolding of the two independent but co-occurrent hazards, and the second one is upgraded to capture the shifts in vulnerability by enriching it with additional element and connection types.

This research work makes a significant contribution to the field of DRR by broadening the purpose of the Impact Chain, transforming it into a first-hand, semi-qualitative tool for analysing vulnerability. Through this expansion, the Impact Chain is elevated from a documentation tool to a diagnosis and prediction instrument. The focus is on advancing its application to delve into the intricate multi-hazard impacts, along with their ramifications on vulnerability conditions. The conceptual framework dwells on the argument of Otto and Raju (2023), who highlight that climate change should not be entirely blamed for climate-related disasters and that vulnerability conditions must be factored in when analysing impactful events. Placing greater emphasis on the vulnerability component brings up the necessity of understanding its dynamics across time and space (de Ruiter and van Loon 2022), and even more in multi-hazard situations. This can be achieved by expanding the scope of Impact Chains to give visibility to such shifts in vulnerability, to diagnose past or present multi-hazard risk management, and to predict potential crises, shortcomings of management approaches, and the transformation of certain vulnerabilities into drivers of vulnerability.

## 2 Setting the scene

### 2.1. Flood risk and hazardous events in Romania

During the first two years of the pandemic, hydro-climatic hazardous events stood out in terms of both powerful impacts and co-occurrence with the COVID-19 pandemic, with almost 800 worldwide hazardous events in January 2020-July 2022 (EM-DAT 2022). Floods have been recognised as the most frequent and destructive natural hazards before and after the pandemic, since 80-90% of all documented disasters associated with natural hazards were determined by floods in the last decade (WHO 2020a) and 36.77% of the worldwide disasters that involved natural hazards in the aforementioned period were related to flood events (EM-DAT 2022).

Climate change is expected to further increase flood frequency and intensity (Dankers and Feyen 2008, Alfieri et al. 2015), and also flood risk (Hettiarachchi et al. 2018). Therefore, floods are bound to become even more impactful given their amplification under climate change, seconded by population growth (Swain et al. 2020) which is closely intertwined with the economic development of flood-prone areas (Tanoue et al. 2016). Climate change will guide societal dynamics (including vulnerability conditions) in the years to come, having a substantial effect on the lives and well-being of future generations, but it will also foster the occurrence of multi-hazard disasters, with grim implications, especially during COVID-19 pandemic waves or future pandemics (Phillips et al. 2020).



The range of natural hazards that affect Romania (i.e., earthquakes, landslides, floods, droughts, cold and heat waves, and
blizzard) is conditioned by its geographical position, its geologic, geomorphologic, climatic, and hydrologic settings. Among
these, floods are the most common and one of the most impactful natural hazards, causing significant damage throughout the
country. The EM-DAT (2023) database includes 102 natural hazardous events that occurred in Romania in 1900-2023, of
which flood events represent almost 52%. These floods resulted in more than 1700 deaths, more than 146600 homeless people,
over 1.64 million affected people, and total estimated damages of about 8.69 billion dollars. This incomplete dataset,
complemented by other European flood-related databases (e.g., HANZE v2.1 developed by Paprotny and Mengel 2023,
Paprotny et al. 2023) points out the prominence of floods among the natural hazards that occur in the country of reference. In
the history of Romania, 1970, 1975, 1991, 2005, 2006, 2008, and 2020 (Chendes et al. 2015, Romanescu et al. 2017, Zaharia
and Ioana-Toroimac 2018) are marked as years with extreme flood events that caused havoc at local or regional scale, followed
by a challenging recovery process.

Paprotny et al. (2023) highlight that river floods account for about 75% of the number of flood events that occurred in Romania
in 1870-2020. Usually, river floods follow a seasonal pattern, with the largest events occurring in the late spring months and
early summer months due to the convergence of high rainfall amounts and snow melting in mountainous areas. This water
input increases the discharge of both main rivers (e.g., the Danube, Siret, Prut, Olt, Mureș, and Argeș rivers) and low-rank
streams. This high level of flood hazard overlaps long-standing vulnerability conditions that are only partially discussed in the
scientific literature (Constantin-Horia et al. 2009, Constantinescu et al. 2015, Vinke-de Kruijf et al. 2015, Peptenatu et al.
2020): deforestation, the extension of the residential areas and transport networks in floodplains and other flood-prone areas
because of inconsistent law enforcement, infrastructure-related failures (e.g., poor-performing, undersized urban sewage
systems), a reactive approach to flood management that neglects the preparedness facts and does not properly understand what
salient recovery involves (Mohammadi et al. 2023). In fact, the last National Synthesis of the Flood Risk Management Plan
(2023) still focuses on generic (i.e., forest and bridge-related measures, about 50%), structural methods (about 33%) to reduce
the flood risk at national scale and also includes confusions between risk-related terminology (e.g., exposure, hazard,
vulnerability).

The significant flood hazard and vulnerability levels result in a high flood risk that materialises once every few years into very
impactful flood events. The flood risk is addressed by the Flood Risk Management Plans elaborated for the 11 Basinal
Administrations that function at county scale. On a national scale, flood risk management is coordinated by several
organisations: the Ministry of Environment, Water and Forests, the National Administration of Romanian Water, and the
National Institute of Hydrology, and Water Management. These organisations are often criticised for their underperformance
in managing flood risk by both scientists (Vinke-de Kruijf et al. 2015) and civil society; an attitude which is justified by the
wrecked aftermath of large flood events that were forecasted and communicated by hydrological warnings.

Flood risk management is not sufficiently documented in Romania, as demonstrated by the lack of databases regarding the
occurrence and impacts of floods. Such information has to be obtained from alternative sources, like weather and hydrological
forecasts and news reports. The flood events taken under analysis in this paper were identified using the hydrological warnings



issued by the National Institute of Hydrology, and Water Management during 2020 and 2021, an approach that was introduced by Albulescu (2023).

Figure 1 shows the number of hydrological warnings by code type, with the red code being the most severe. The most numerous and severe hydrological warnings were issued in June-July 2020 and May-July 2021, fitting the seasonal patterns of river floods. The plotting of the warnings offered important clues about the occurrence of powerful flood events, which were corroborated with information extracted from news reports, as described in the Methodology section. The spatial extent of the various impacts of the powerful river floods in 2020 and 2021 is presented in Figure 2, and detailed in the Results.


Figure 1: Number of hydrological warnings issued in 2020 and 2021 in Romania





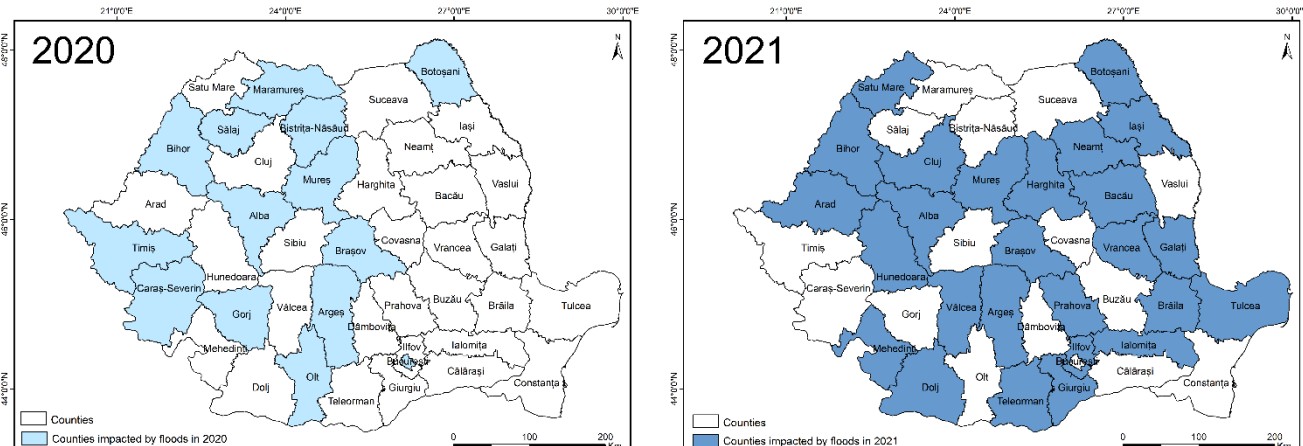

**Figure 2: The Romanian counties impacted by the flood events of 2020 and 2021**

## 2.2. The COVID-19 pandemic in Romania

The first confirmed case of COVID-19 registered in Romania occurred on the 26th of February 2020, and the first two deaths due to this disease occurred approximately a month later. Until the beginning of June 2023, more than 3.4 million cases of COVID-19 and over 68,000 deaths were registered in the country of interest, of which 53.07%, respectively 86.09% can be

traced back to the first two pandemic years (WHO Dashboard 2023). The largest number of both COVID-19 cases (1,179,282) and COVID-19-induced deaths (43,118) occurred in 2021. This human toll unfolded in five pandemic waves (Figure 3), of which the fourth one, starting in 2022, was the most aggressive. This fourth pandemic wave was preceded by a smaller but still high-level one, which concluded in 2021.





**Figure 3: The dynamics of the new cases of COVID-19 in Romania with a highlight on 2020 and 2021, plotted against the periods with/without restrictions and the clusters of flood events (COVID-19 data source: WHO Dashboard 2023)**

Like in many other countries, the pandemic waves in Romania followed a seasonal pattern that was conditioned by temperature and humidity (Mecenas et al. 2020). Figure 3 indicates that the same seasonal pattern was followed by the COVID-19-related restrictions. As an immediate response to the emergence of COVID-19 cases, at the end of March 2020, the Romanian Government declared the National State of Emergency (Decree no. 195/2020) and imposed lockdown, which was severe compared to the one implemented in other counties. This ended on the 15th of May 2020, and was followed by a 2-year





National State of Alert during which periods free of restrictions – that overlapped the summer months, alternated with periods of circulation restrictions for citizens that aimed to tilt the SARS-CoV-2 infection curve – that were specific to the cold season (Figure 3). Another preventive measure worthy of attention, implemented in the early pandemic months (i.e., March-August 2020) was the mandatory hospitalisation of COVID-19 positive patients, regardless of the presence of symptoms; which resulted in additional pressure and challenges for the Romanian medical system. The mandatory hospitalisation, together with the quarantine and isolation of infected patients, were declared unconstitutional by the Romanian Constitutional Court (CCR 2022), fuelling the discontent of Romanian civil society and stimulating the crumble of its trust in national authorities (Džakula et al. 2022).

The COVID-19 vaccination campaign that started in December 2021 introduced a new variable to be taken into account when establishing circulation restrictions: whether people were vaccinated or not. At the end of May 2021, unvaccinated citizens were subject to curfews that did not apply to vaccinated ones were also being prohibited from joining gatherings. These restrictions were lifted for all citizens at the end of June 2021, only to be implemented again towards the end of the year. For instance, from the end of October 2021 until March 2022, unvaccinated people were banned from circulation between 22 p.m. and 5 a.m., from social gatherings, and from entering large commercial centres, restaurants, hotels, or entertainment facilities unless proving that they were not infected with the COVID-19 virus with the result of a test.

Figure 3 shows that the flood events that occurred in June 2020 correspond to the beginning of a restriction-free period, which was followed by one with severe restrictions. The floods of January 2021 overlapped a period with restrictions for everyone, when wearing face masks was mandatory, circulation was prohibited between 23 p.m. and 5 a.m., social gatherings were banned, and a large part of work was moved to the virtual environment. Towards May 2021, circulation restrictions were lifted only for vaccinated people, and it was not until the 26th of July that all COVID-19-related restrictions ceased; this means that the flood events that happened on the 13th and 18th of May overlapped a period with restrictions for unvaccinated people and that the ones in June-August correspond to a restriction-free interval. The flood events of December 2021 occurred during a period of restrictions imposed on unvaccinated people.

## 3. Methodology

The proposed methodological framework aims to identify and analyse the multi-hazard impacts, along with changes in vulnerability conditions within a multi-hazard context. This framework dwells on Impact Chains as instruments for documentation, visualisation, organisation, and scientific inquiry, ultimately broadening their application to fit the latter objective of studying the dynamics of vulnerability – particularly the augmentation of vulnerability, and turning them into diagnosis and prediction tools. This section presents three distinct workflows in the methodological framework (Figure 4); briefly introducing and explaining the building of the Impact Chain, proceeding with its exploration, and emphasising its enhancement.





Natural Hazards
and Earth System

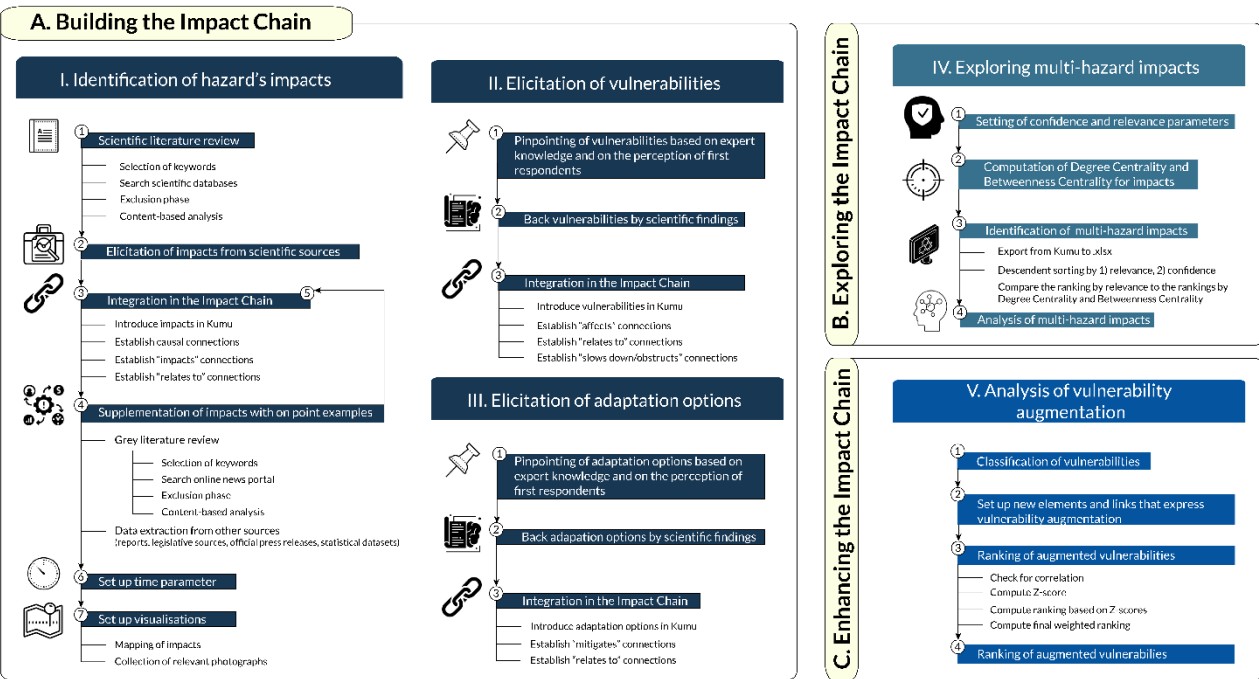

**Figure 4: Methodological framework**

## 3.1. Building the Impact Chain

Impact Chains represent conceptual models designed to facilitate the investigation of climate and disaster risk under a
structured analysis framework for the risks associated with climate-related impacts (UNDRR 2022). They have been used for
elicitation, conceptualisation, analysis, and information sharing purposes, as tools that explore and analyse the impacts of
single hazards or multi-hazards specific to past or potential hazardous events, following different operational frameworks (e.g.,
expert workshop, desktop analysis, machine-generated) and taking into consideration different spatial and temporal scopes
(Pittore et al. 2023).

The structure of an Impact Chain includes elements that can be considered the fundamental units of a hazard-related context
and the connections established between them. These elements can take the form of hazards, impacts, exposed elements,
vulnerabilities, or adaptation options. They are organised in a chain-resembling structure that relies on different connection
types: causes, affects, relates to, impacts, and mitigates. Detailed guidelines on how to build such structures are provided by
Pittore et al. (2023).

In this paper, Impact Chains are used as methodological instruments that help to document, visualise, organise, and analyse
the outcomes of multi-hazard events, following an expert-based desktop analysis that is underpinned by empirical evidence



collected from multiple data and information sources, including scientific literature and grey literature. Here, we extend the application of Impact Chains to explore the third type of vulnerability dynamics identified by de Ruiter and van Loon (2022), namely the changes in vulnerability conditions related to compounded hazards, or more accurately, the augmentation of vulnerability in a multi-hazard context. Elevating the Impact Chain from its above mentioned original purposes to a diagnosis and prediction tool represents a pioneering research endeavour, standing out as an element of methodological novelty.

Such efforts are vital for elaborating post-pandemic update risk management plans that avoid inadvertently introducing additional sources of unforeseen vulnerability. Risk (or hazard) management can act on vulnerability conditions both ways (de Ruiter and van Loon 2022): producing desirable results (i.e., by decreasing vulnerability) or unwelcome outcomes (i.e., by augmenting vulnerability). The literature provides examples where the risk management of a certain hazard was responsible for increasing the risk associated with another hazard (Ward et al. 2020, de Ruiter et al. 2021a, b); and there are fair chances that this will happen again if the dynamics of vulnerability in multi-risk situations is not properly understood.

Figure 4 illustrates a comprehensive three-part breakdown of the construction of the Impact Chain through a combination of knowledge, data and information extracted from a diverse range of sources: scientific papers, legislative documents, official press releases, reports, statistical datasets, and grey literature in the form of news reports (Figure 4). The Impact Chain was implemented in Kumu, which is a powerful mind mapping tool that allows for a variety of mapping settings (e.g., stakeholder, systems, social network, community asset, concept mapping), as well as import and export options (Kumu 2023).

The first phase of the building process (A in Figure 4) relied on a literature review regarding the impacts of flood events and the pandemic, complemented by a grey literature review that provided on-point examples. As part of the first review, the most prominent scientific databases (i.e., Web of Science, Google Scholar, ResearchGate, and PubMed) were searched for relevant papers using the following keywords: "Covid-19 pandemic Romania", "Covid-19 pandemic impact Romania", "Floods Romania 2020", "Floods Romania 2021". Next, during the exclusion phase, the titles and abstracts of the collected articles were analysed in order to select only the research works with a clearly defined and relevant aim, a thorough and methodologically validated analysis of the impacts of the hazards, and an adequate spatial and temporal focus. In the last phase of the literature review on impacts, content analysis was performed on the selected papers, and the relevant impacts were included in a database. The grey literature review was performed using a prominent online Romanian national news portal Digi24 (2023). It was limited to the impact of extreme floods and did not include the impacts of the pandemic. These were extracted from legislative documents (Decree no. 195/2020), official press releases (CCR 2022) or reports (WHO 2020b, HSRM 2021a, b, OECD 2021, CDC 2022, WHO Dashboard 2023), and statistical datasets (Eurostat 2021).

The last two phases of the construction process focused on the elicitation of vulnerabilities and adaptation options under an expert knowledge-based approach. The identification of these elements was backed by scientific findings wherever possible in order to obtain a valid configuration of the chain (A in Figure 4).

An addition to the Impact Chain developed in the early stages of the Paratus Project (PARATUS Deliverable 1.1 2023) was to integrate the feedback of 595 first responders involved in flood management in 2021, focusing on aspects concerning preparedness, coordination, and experience, upon extracting them from the study of Fekete et al. (2023). Their perception of





the problems encountered during flood-related emergency interventions, potential improvements, cooperation among volunteers, provision of information about the deployment, and flood-affected infrastructure served as a basis for eliciting a new set of vulnerabilities and adaptation options.

Regardless of their type, all elements and connections were integrated into Kumu with a short description, associated sources, and references. The Kumu design for appropriate flood impacts was enriched with photographs and maps depicting the spatial distribution of impacts in 2020, 2021 or both, at county or local scale. Cumulatively, the Impact Chain drew from 46 scientific papers (including one on the feedback of first responders), one legislative document, one official press release, one Eurostat statistical dataset, 6 official reports, and 75 news reports. All the connections in the Impact Chain, regardless of their type or the elements they connect, were described and assigned values for the Sources or References parameters.

**3.2. Exploring the multi-hazard impacts**

The exploration of the Impact Chain (B in Figure 4) revolves around the objective of identifying multi-hazard impacts, defined as the intertwined effects of compounded hazards that affect the same area in the same time period (Zscheischler et al. 2018, Tilloy et al. 2022). In other words, the impact of one hazard affects the manifestation and outcomes of the second co-occurring hazard, leading to complex and harder-to-manage multi-risks.

To this end, the Impact Chain was completed by setting up two more apposite parameters for each element and connection: relevance and confidence. The first refers to the significance of a given element or connection to the scope of the Impact Chain, while the latter designates the reliability of the information conveyed by the element or connection. While the guidelines of Pittore et al. (2023) are most informative of the elements and connection types, they leave the way open for setting up customised rules for relevance and confidence.

For the proposed Impact Chain, the authors assigned values ranging from 1 to 10 to both parameters, with the largest numbers accounting for the highest relevance and confidence. The process of attributing these values was guided by questions that varied from elements to connections. These guiding questions and the rules illustrated in Figure 5 represent another element of novelty included in the Impact Chain developed in this study.




**Relevance of elements**

**Guiding question**: *How important/relvant is this element for both floods and the COVID-19 pandemic?*

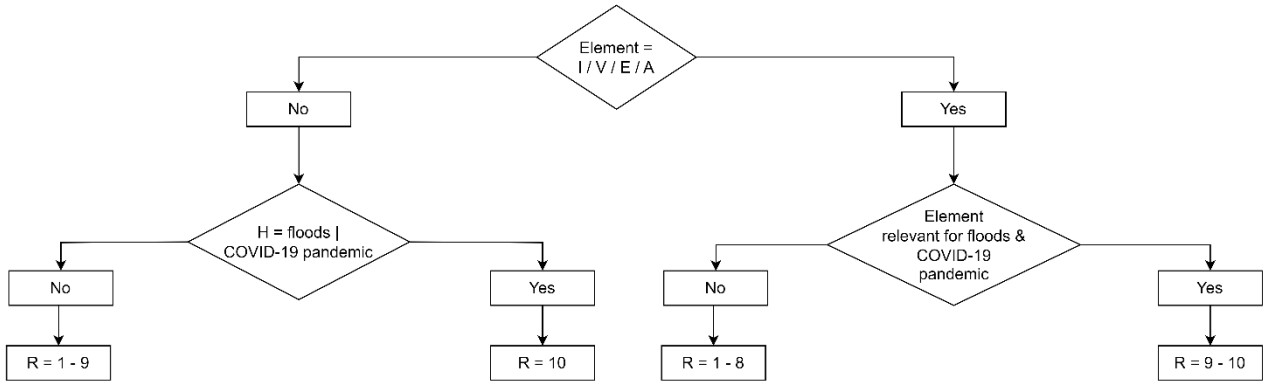

**Relevance of connections**

**Guiding question**: *How important/relvant is that the X element \*causes/affects/impacts/mitigates/relates\* to the Y element, for both floods and the COVID-19 pandemic?*

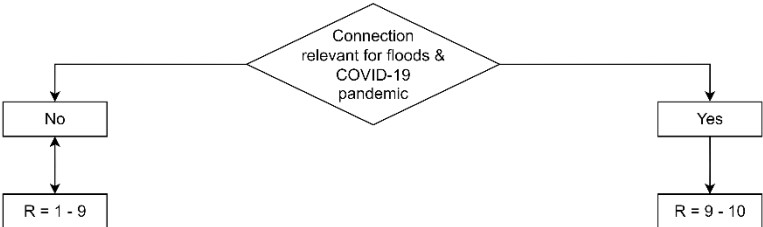

**Confidence of elements and connections**

**Guiding questions**: *What is the type of the source(s)? How many sources are considered?*

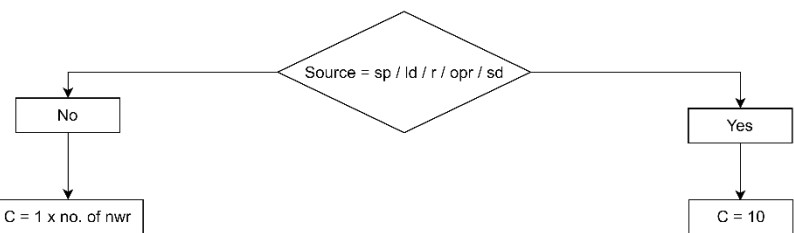

**Figure 5: Logical data model for setting the values of relevance (R) and confidence (C) of elements and connections in the Impact Chain. Elements: H – hazard, I – impact, V – vulnerability, E – exposed element, A – adaptation option. Source types: sp – scientific paper, ld – legal document, r – report, opr – official press release, sd – statistical dataset, nwr – news report. Logical operators: & – and, | – or. Where multiple different sources can be attributed to the same element or connection, the confidence was established at the maximum value of any of the sources.**

Upon setting the relevance and confidence of the impact elements, the multi-hazard impacts were identified based on the highest values of these parameters, combined with two other metrics computed automatically in Kumu. Out of the range of metrics used to analyse the Impact Chain (Table 2), the Degree Centrality and Betweenness Centrality were selected for the



purpose of identifying the most prominent and most connected impacts. Their values were sorted in reversed order, which put the most connected and bridged impacts at the top of the list. These top elements were the most likely to be included in multi-
hazard impact pathways and therefore should have obtained the highest relevance scores.

**Table 2. Metrics of Kumu that were used to analyse the Impact Chain**

| Metrics | Description | Source |
|---|---|---|
| Degree Centrality | Degree centrality is the simplest of the centrality metrics, counting the number of connections an element has. In general, elements with a high degree are the local connectors/hubs, but aren't necessarily the best connected to the wider network. | Kumu (2023) |
| Betweenness Centrality | Betweenness centrality measures how many times an element lies on the shortest path between two other elements. In general, elements with high betweenness have more control over the flow of information and act as key bridges within the network. They can also be potential single points of failure. | |

### 3.3. Enhancing the Impact Chain

The extension of the Impact Chain capabilities to identify the vulnerability increases induced by impacts resulting from single
or multiple hazards or by associated adaptation options (C in Figure 4) constitutes an innovative element, which elevates the tool's investigative prowess and practicality. This broadening of the original application of the Impact Chain was done by 1) introducing new types of elements (i.e., augmented vulnerabilities, derived impacts), 2) establishing new types of connections between the impacts/adaptation options and vulnerabilities, and 3) ranking the vulnerabilities in the Impact Chain based on their augmentation. These steps were implemented to construct an enhanced Impact Chain, building on the previous version
that documented the unfolding of the selected co-occurrent hazards in Romania in 2020-2021. With this addition, the documentary focus of the chain progresses to a more analytical stance, specifically geared towards diagnosing multi-hazard management  and predicting potential crises, deficiencies in management approaches and the transformation of certain vulnerabilities into drivers of vulnerability.

The first step was to perform an in-depth analysis of the vulnerabilities and their already established links in the initial Impact
Chain. The vulnerabilities were grouped according to their related hazard, type, spatial scale, and links to specific adaptation option(s). This classification provided a better understanding of the contribution of vulnerabilities to the manifestation of flood and COVID-19 pandemic impacts.

Further on, the Impact Chain was enhanced by introducing new connection types between the impacts/adaptation options and the vulnerabilities (Figure 6), drawing from the types of maladaptation to climate change and their implications on vulnerability
proposed by Schipper (2020). The three types of maladaptation in question (i.e., rebounding vulnerability, shifting vulnerability, and creating negative externalities) were adapted to the multi-hazard context and complemented by a new connection type also relevant to the Impact Chain (i.e., deepening vulnerability). These new connections account for the





augmentation of a vulnerability by a given impact or adaptation option in a way that could not have been prevented or precluded. The new links are defined as:

•     Deepens (vulnerability): the augmentation of a vulnerability by an impact, both relating to the same hazard;

        •     Shifts (vulnerability): the augmentation of a vulnerability to a certain hazard by an impact caused by a different hazard;

        •     Rebounds (vulnerability): the augmentation of a vulnerability by an adaptive option that aimed to attenuate an impact but ended up increasing a vulnerability;

•     Creates negative externalities: the augmentation of a vulnerability by an adaptive option that has adverse effects on anyone who was not targeted by it (Schipper 2020).

To set up the new connections, each impact in the initial Impact Chain was studied from the perspective question of "Which vulnerability can be augmented by this impact?". The adaptation options were also scanned according to the same adapted question, with the goal of identifying possible unwanted effects of measures intended to lessen certain vulnerability conditions,

as reported by de Ruiter et al. (2021a, b). The vulnerabilities that were connected with impacts and/or adaptation options through the above said new links that express vulnerability augmentation were transformed into elements called augmented vulnerabilities.

A noteworthy situation that emerged from the experience-based feedback of first responders is the one where certain vulnerabilities that influence the manifestation of impacts can also slow down or obstruct the implementation of adaptation

options. Such instances were marked by a new type of connection called "slows down/obstructs", established between vulnerabilities and adaptation options.

Within the new conceptual framework of the enhanced Impact Chain (Figure 6), certain augmented vulnerabilities stand out also as impacts that deepen the impact that increased the vulnerability in the first place. Such augmented vulnerabilities that also act as impacts were introduced in the enhanced Impact Chain as derived impacts and linked to the vulnerability element that they share their name with by "relates to" connections. These "relates to" links are not visible within the enhanced Impact

Chain in Kumu in order to reduce the visual strain. Subsequently, the derived impacts were linked with the impact that augmented the corresponding vulnerability by a new type of connection named "sharpens" (Figure 6). The "sharpens" connections convey the message that the augmented vulnerability reflects back on the impact that increased it, making it even more prominent than in the beginning.







**Figure 6: Conceptual framework of the new elements and links of the enhanced Impact Chain**

The ranking of the vulnerabilities based on their augmentation relied on the number of augmentation connections from impacts to vulnerability (i.e., deepens, shifts) and on the number of augmentation connections from adaptation options to vulnerability (i.e., rebounds, creates negative externalities). These were computed for each of the 26 vulnerabilities in the enhanced Impact Chain and subsequently checked for Pearson correlation. The absence of correlation allowed for the computation of the Z-score. This score indicated the extent to which each vulnerability deviated from the average in terms of standard deviation. Next, two rankings of the vulnerabilities were calculated based on the Z-scores of the augmentation connections of impact-vulnerability and adaptation option-vulnerability. The final ranking was computed using an expert-based weighted approach, by attributing the impact-vulnerability connections a weight of 70% and the adaptation option-vulnerability connections a weight of 30%. The rationale behind the assigned weights lies in the observation that impacts augment vulnerabilities to a greater extent than adaptation options do, at a ratio of 53.84% augmentation by impacts vs. 3.84% augmentation by adaptation





options, while the remaining 11.53% is attributable to both impacts and adaptation options combined. The ascending order of the final ranking showed the extent to which the vulnerabilities were overall augmented, from the most to the least augmented.

## 4. Results

This section focuses on the analysis of the multi-hazard impacts, with special attention placed on the logical pathways that involve impacts of both hazards and on the fluctuations of vulnerability as a result of different impacts. The analysis of multi-hazard impacts is performed using the initial Impact Chain (included in PARATUS Deliverable 1.1 2023), while the variations in vulnerability are examined within the enhanced Impact Chain. The differences between these two chains are outlined in Table 3 and Figure 7. The intricate configuration of the two Impact Chains does not allow for a proper visualisation within

this paper, but they can be accessed online on the Kumu platform using the links in Table 3. To address the convoluted aspect of the initial chain, the enhanced version was restructured to provide a more intuitive and easily manageable visualisation.

**Table 3. Details on the initial and enhanced Impact Chains**

|  | Initial Impact Chain | Enhanced Impact Chain |
|---|---|---|
| **Focus** | Documentative focus | Diagnosis and prediction focus |
| **Types of elements** | Hazard, Impact, Vulnerability, Adaptation option, Exposed element | Primary hazard, Secondary hazard, Impact, Vulnerability, Augmented vulnerability, Adaptation option, Exposed element, Derived impact |
| **Number of elements** | 81 | 102 |
| **Types of connections** | Causes, Impacts, Affects, Mitigates, Relates to | Causes, Impacts, Affects, Mitigates, Relates to, Slows down/Obstructs, Deepens, Shifts, Rebounds, Creates negative externalities, Sharpens |
| **Number of connections** | 211 | 312 |
| **URL** | Initial Impact Chain URL | Enhanced Impact Chain URL |
|  | Reference list | |




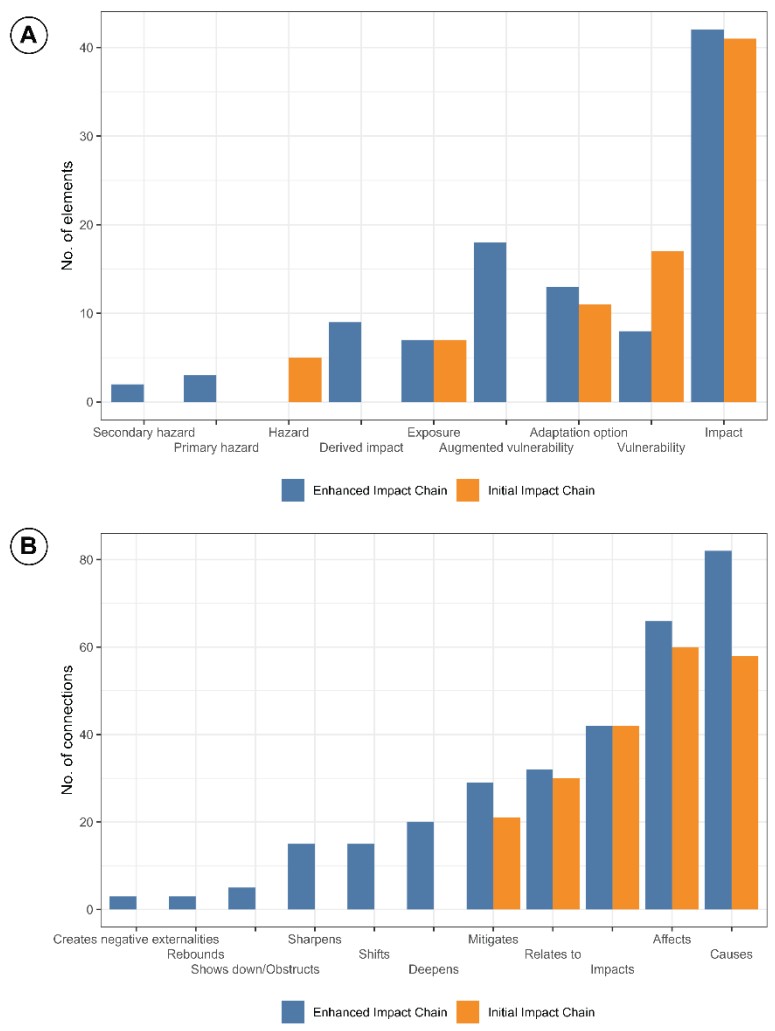


**Figure 7: The number of A. elements and B. connections in the initial and enhanced Impact Chains**

## 4.1. Multi-hazard Impacts

Floods, the COVID-19 pandemic, and heavy rainfall were considered primary hazards within the Impact Chain, but only the first two are analysed in this study due to their significant impacts. Other secondary hazards (e.g., strong wind, landslides) co-
occurred with the other two, but their role was of lesser significance in the multi-risk context. In 2020, there were five major flood events that imposed the evacuation of people, all in June (16th, 18th, 19th, 23rd, and 26th). In the subsequent year, there were 8 such events, of which two occurred in May (13th, 18th), two in June (18th, 19th), and four in July (15th, 16th, 19th, 20th) (Albulescu 2023). In addition, flood events that did not involve evacuation procedures but were still included in the Impact Chain, occurred in January, August, and December 2021. Only the 2020 flood events resulted in three human casualties
(Albulescu 2023), which is confirmed by the HANZE v2.1 (2023) database. The human toll of the other primary hazard (i.e.,





the COVID-19 pandemic) was much larger: 15,596 deaths in 2020 and 43,118 deaths in 2021. These represent about 86% of the total COVID-19 casualties registered in Romania across all 5 pandemic waves (WHO 2023). By the end of 2021, Romania had been affected by three pandemic waves, the last one being one the most aggressive (Figure 3).

The extreme flood events and the pandemic had particular sets of impacts: flooded business buildings, including tourism accommodation, disruption of tourism activities, dead or missing animals, flooded croplands, damaged/destroyed assets (e.g., cars, furniture, electronics), damaged bridges, river water contamination with garbage resulting from the flood events; and economic challenges, increased unemployment, work overload on medical personnel, decreased life expectancy, changes in work patterns,  in the case of the pandemic. The spatial extent of the flood impacts in each of the analysed years is shown in Figure 8.

Some of the impacts of the two primary hazards interact in various ways and to different extents, forming the foundation of multi-hazard risk and representing the primary focus of the following analysis. Table 4 shows the top 10 impacts by Degree Centrality and Betweenness Centrality computed for the initial Impact Chain. The elements that emerge when considering both of the metrics are the flooded/damages/blocked roads, the flooded/damaged households or houses, the potential increase in the COVID-19 new cases, and road transportation impairment. As expected, all of these were assigned maximum relevance

scores. The systematic evaluation of impacts, incorporating two Kumu metrics in conjunction with the relevance parameter, grounded the identification of pathways relating to both hazards. This represents one of the contributions to the Romanian Case Studies presented in PARATUS Deliverable 1.1 (2023).




**Figure 8: Spatial extent of the impacts of the extreme flood events that affected Romania in 2020 and/or 2021. Impacts: A – Human casualties, B – Displaced/(Self-) Evacuated people, C – Flooded/Damaged households or houses, D – Damaged bridges, E – Isolated human communities, F – Railway transportation impairment, G – Damaged facilities/Cut off of electricity/gas/water supply, H – Sewage system overflow, I – Fallen trees, J – Landslides, K – River water contaminated with garbage, L – Dead/Missing animals, M – Flooded croplands, N – Damaged cars, O – Disrupted tourism activities, Q – Flooded business buildings, P – Flooded public buildings (including 1 hospital), R – Distrupted ambulance service**



**Table 4. The impacts ranked by Degree Centrality and Betweenness Centrality, from the initial Impact Chain**

| Impact | Degree Centrality | |
|---|---|---|
| | **Value** | **Ranking** |
| Flooded/Damaged/Blocked roads | 16 | 1 |
| Flooded/Damaged households or houses | 16 | 1 |
| Economic loss | 12 | 2 |
| Potential increase in the COVID-19 new cases | 11 | 3 |
| Lockdown | 10 | 4 |
| Road transportation impairment | 9 | 5 |
| Flooded croplands | 8 | 6 |
| Railway transportation impairment | 8 | 6 |
| Increased hospitalisation costs | 7 | 7 |
| Human casualties | 7 | 7 |

| Impact | Betweenness Centrality | |
|---|---|---|
| | **Value** | **Ranking** |
| Flooded/Damaged/Blocked roads | 0.015 | 1 |
| Flooded/Damaged households or houses | 0.009 | 2 |
| Temporary disruption of COVID-19 vaccination centres | 0.006 | 3 |
| Fallen trees | 0.005 | 4 |
| Road transportation impairment | 0.005 | 4 |
| Sewage system overflow | 0.003 | 5 |
| Potential increase in the COVID-19 new cases | 0.003 | 5 |
| Health problems | 0.003 | 5 |
| Damaged lifelines | 0.003 | 5 |
| Displaced/(Self-) Evacuated people | 0.003 | 5 |

A notable pathway refers to the "pushing" effect of the flood events, which were responsible for setting both people and resources into motion, increasing the probability of SARS-CoV-2 infection. Firstly, the flood events in 2020-2021 resulted in widespread damage to hundreds of houses and households across the country, with the greatest impact in the Western and

North-Western regions of Romania (Figure 8C). The houses and households were invaded mostly by river water, but in urban areas, the source was represented by the overflows from the faulty sewage systems. Certain deeply rooted vulnerabilities contributed to this outcome, among which the position of households at short distances from rivers, the low quality of construction materials, the improper governance structure, and insufficient or ineffective hard engineering infrastructure/measures are worth mentioning. All of these vulnerabilities stem from the trend of extending inhabited areas

into flood-prone areas.

People could no longer inhabit their severely damaged houses, which means that they had to be evacuated and displaced, or, in cases of imminent danger, they self-evacuated (Figure 8B). In 2020, it is estimated that 340-720 people were evacuated because of powerful floods, and the total number of affected people reached 1550 (HANZE v2.1 2023). The next year, the





number of evacuees exceeded 675 at national level (Albulescu 2023). Evacuees were provided with temporary accommodation in makeshift emergency shelters, some of which were set up in local cultural buildings or nearby indoor sports venues. In other cases, the evacuated people found accommodation at their relatives or at neighbours who were not affected by floods. These flood-determined gatherings favoured the chances of SARS-CoV-2 infection, which concludes the path to a multi-hazard impact. The COVID-19 preventive measures implemented during the evacuation process or inside emergency shelters are not documented. It is possible that flood management was not calibrated to pandemic requirements, which means that classical protocols did not account for the epidemiological conditions. This represents a top-level vulnerability, especially when considering the potential effects of flood events on the dynamics of the new confirmed cases at county level studied by Albulescu (2023). In almost all counties (with one exception) that were severely affected by the flood events of 2020 and 2021, and where evacuation procedures were performed, the number of new cases increased after 14 days (i.e., the extended incubation time of SARS-CoV-2) since each flood event. The largest increase was 208 new cases, but most of the increases did not exceed 50 new cases (Albulescu 2023).

Another prominent multi-hazard impact starts with damaged infrastructure and flood-determined transport impairment. In both reference years, during and immediately after the extreme flood events, sections of national or county roads were damaged or blocked by fallen trees, or covered by flood water or sediments transported by rivers. In some cases, large portions of roads were covered by water all together, be it river water or sewage system water (in Bucharest, Arad, Brezoi, Craiova, Galați). In addition, there were several instances where bridges were destroyed or damaged, leaving parts of rural settlements or small human communities isolated for a few hours or days (Figures 8D, 8E). The described damaged infrastructure impeded or at least hindered road transportation, with additional implications for the local management of the pandemic or medical emergencies. For instance, in June 2021, the activity of the vaccination centre in Suraia, Vrancea County, was put on hold for several days after the Putna River destroyed the levee near Biliești village and county road 204D was covered by water. Thus, the unfunctional road prevented both medical personnel and citizens from accessing the centre, which delayed the vaccination process with uncertain (but still concerning) implications for the dynamics of COVID-19 infection in the proximal rural communities. More clear-cut consequences emerge in the case of the flooding of a road section in Argeș County in June 2020, which prevented an ambulance from reaching a patient in need of health care because of backbone problems. Behind these impact examples are long-lasting vulnerabilities: the development of infrastructure in flood-prone areas (that could also have been deforested), which converges with the low-quality of construction materials used to repair or extend roads and even insufficient or ineffective hard engineering infrastructure or protective measures.

Continuing along the line of flood-determined infrastructure damages, the cut-off of electricity/gas/water supply that occurs during or immediately after powerful flood events (because of fallen trees that damage the lifelines, for example) holds the potential to disrupt the functionality of medical equipment. Power outages occurred throughout Romania in the aftermath of the extreme flood events of 2020-2021, with no reported consequences for the hospital electricity networks. Nevertheless, should such a blackout occur in a major urban centre during a pandemic wave, the outcome for the intubated COVID-19 patients could be fatal.





An additional distinctive multi-hazard impact that also relates to faulty infrastructure starts with the frequent overflow of sewage water in urban areas such as Bucharest, Arad, Brezoi, Craiova, and Galați cities. Typically, such overflows occur in poor neighbourhoods (in Craiova and Galați cities), thereby affecting the most vulnerable communities. These neighbourhoods lack proper urban infrastructure in general, but the vulnerability that contributes to the stated impact (i.e., the undersized, outdated, and ineffective sewage systems) prevails in the entire city. Sewage system overflow results in the flooding of houses, households, and roads, which in turn may determine the evacuation of people and water contamination, with potential implications for human health. These implications may even be related to the COVID-19 pandemic, since Han and He (2021) argue that exposure to sewage water might facilitate the transmission of the SARS-CoV-2 virus. However, these findings have to be conclusively established.

It should be highlighted that not only households and residential buildings were flooded but also business buildings and public institution buildings, including a ward of one hospital. The basement of the orthopaedic ward in Timișoara City was flooded in June 2020, with potential negative consequences for its functionality. Whether flood water damaged the electricity network or medical equipment in the hospital remains uncertain in this case, but such multi-hazard impacts should not be excluded from the picture in the future. In such instances, the flooding of hospital wards shifts the focus of medical staff from providing adequate care to patients in need, to solving the pressing issue of removing flood water and keeping the medical equipment functional. The flooding of hospital buildings can temporarily hinder health care provision at best, or even stop the functioning of life support equipment at worst; which can have fatal consequences during pandemic waves, when the ICUs work at full capacity.

### 4.2. Analysis of vulnerability augmentation

### 4.2.1. Classification of vulnerabilities

The enhanced Impact Chain was upgraded from 17 to 26 vulnerabilities upon integrating the perception of first responders. More than half (57.7%%) of these vulnerabilities were related to flood events, 23.1% pertained to both hazards, and 19.2% of them to the pandemic (Figure 9A). Most of the vulnerabilities contribute to prominent multi-hazard impacts such as the flooded/damaged houses or households, the flooded/damaged/blocked roads, the displaced/(self-) evacuated people, increased stress or anxiety, and the potential increase in COVID-19 new cases.

The vulnerabilities were grouped according to their type, as described in Appendix A. More than a third (34.6%%) of them stemmed from failures of emergency management, while 19.2% of them derive from failures of territorial planning or of medical management (Figure 9B). At the same time, the number of vulnerabilities associated with coping capacity (15.4%) or infrastructure (11.5%) was rather low. In terms of scale, most vulnerabilities were identified at local scale (69.2%), and only 23.1% were specific to the entire country (i.e., national scale) (Figure 9C).





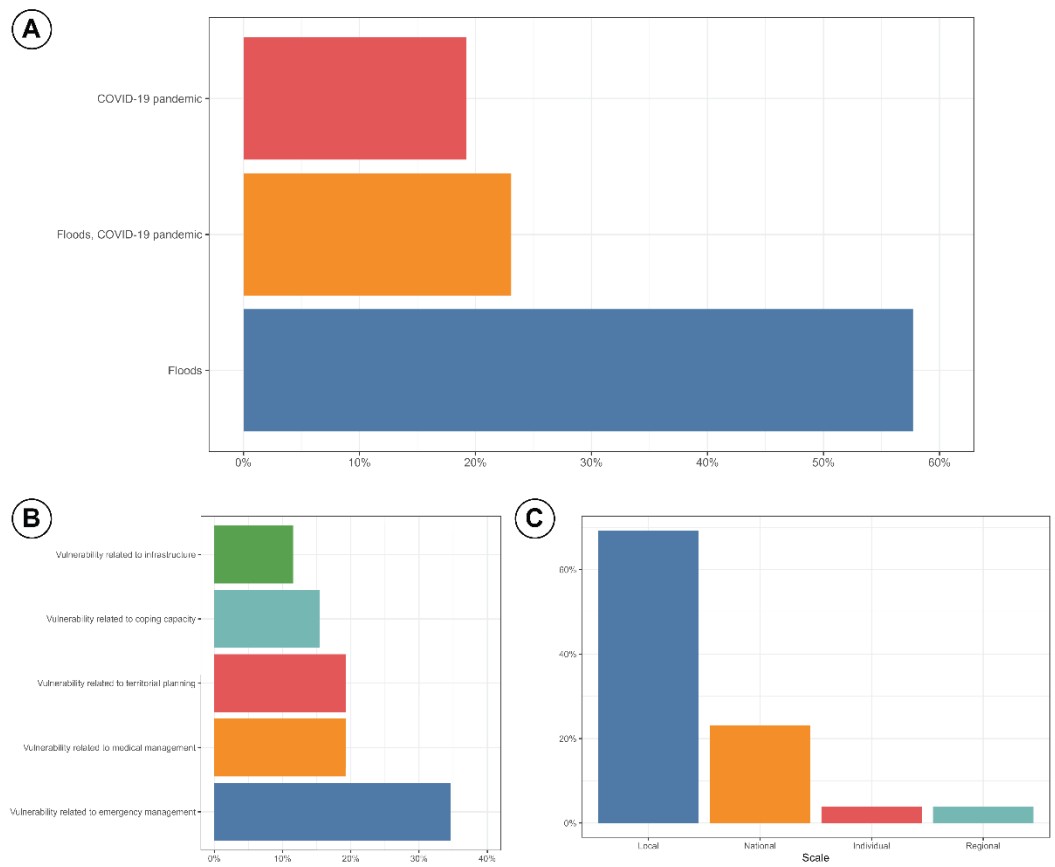


**Figure 9: Proportion of vulnerabilities by A. hazard, B. type, and C. scale in the enhanced Impact Chain**

When it comes to adaptation options, only 30.76% of the vulnerabilities were mitigated by such elements, 3 of them related to the COVID-19 pandemic (i.e., low-performance medical system, insufficient medical personnel, insufficient ICU capacity), the other 3 to both hazards (i.e., flood management not adapted to the COVID-19 context, ineffective institutional

communication, uncooperative population), and the rest to floods (i.e., improper mapping and visualisation of affected areas, lack of equipment for first responders).

The 30% rate of mitigated vulnerabilities      shows that most of the adaptation options targeted impacts, which means that they produced short-term positive change, addressing only to a limited extent the causes of the medical crisis and the multi-hazard vulnerabilities, and even to a lesser extent the flood vulnerabilities. The adaptation options that mitigated vulnerabilities

related to the COVID-19 pandemic were the most numerous: 4 in the case of insufficient ICU capacity, 3 in the case of insufficient medical personnel, and 2 in the case of the low-performance medical system. The main adaptation options related to support from other states (e.g., medical equipment and staff), the transfer of COVID-19 patients to other countries, the establishment of new modular hospitals, and the hiring of additional medical personnel – all of which allowed the fight against the pandemic to continue.



All of the other mitigated vulnerabilities were addressed by a single adaptation option, showing a unilateral approach. In the case of floods, both vulnerabilities were mitigated by an "umbrella" adaptation option that includes various actions specific to each context, namely the great capacity of first responders to develop creative solutions in crisis and cope with new challenges. With few exceptions (e.g., RO-Alert SMS messages and hydrological warnings, which are part of early warning systems), most of these flood-related adaptations focused on alleviating the "symptoms" of the local crisis and did not address its root

causes. For instance, during the flood event on the 18th of June 2020, river banks were heightened by firefighters with sand bags to prevent the water from reaching the houses in proximity at Remetea-Pogănici and Obreja (Caraș-Severin County). Other examples of short-term, recovery-related adaptation options are the removal of fallen trees from streets/roads or flood water from households or buildings.

### 4.2.2. Classification of augmented vulnerabilities and augmentation links

While the adaptation options left most of the vulnerabilities unaddressed (69.23%), several of the impacts of the flood events and the pandemic, or the associated adaptive measures, had an amplification effect on certain vulnerabilities. To identify the augmented vulnerabilities, 41 new connections (Appendix B) that express different forms of vulnerability augmentation were established between the impacts that would potentially generate increases in vulnerability (i.e., deepens or shifts vulnerability links) or from the adaptation options with this effect (i.e., rebounds vulnerability or creates negative externalities connections),

under an expert-based approach.

In the enhanced Impact Chain, 18 (69.23%) out of 26 vulnerabilities were augmented, some of them more than once: 14 (53.84%) were augmented by hazard impacts, 1 (3.84%) was augmented by solely adaptation options, and 3 (11.53%) by both impacts and adaptation options. The vulnerabilities that increased because of both elements are: the uncooperative population, flood management not adapted to COVID-19 conditions, and shallow implementation of preventive measures.

The distribution of augmented vulnerabilities among the hazards is unbalanced: half of the augmented vulnerabilities are specific to floods, 27.77% to the COVID-19 pandemic, and 22.22% to both hazards. Also, the augmented vulnerabilities related to medical or emergency management account for 66.66% of the total, and the other 3 categories (i.e., vulnerabilities related to coping capacity, infrastructure, or territorial planning) each account for less than 20%. Most augmented vulnerabilities manifest at local scale (66.66%), and 22.22% of them at national level. Almost all vulnerabilities that were mitigated by

adaptation options were also augmented either by hazard impacts (i.e., lack of equipment for first responders, improper mapping and visualisation of affected areas, low-performance medical system, insufficient medical personnel, insufficient ICU capacity), or both impacts and adaptation options (i.e., uncooperative population). The only mitigated vulnerability that was not also augmented was the ineffectiveness of institutional communication.

Almost half (48.8%) of the new augmentation connections convey a deepening effect on vulnerability elements, and more than

a third (36.6%) augment vulnerability by shifting it from one hazard to the other (Figure 10). The increases in vulnerability caused by adaptation options total about 14.64%, with equal unwelcome effects (7.32%) resulting from rebounding





vulnerability and creating negative externalities. The details on the augmentation of certain vulnerabilities by impacts or adaptation options are provided in Appendix B.

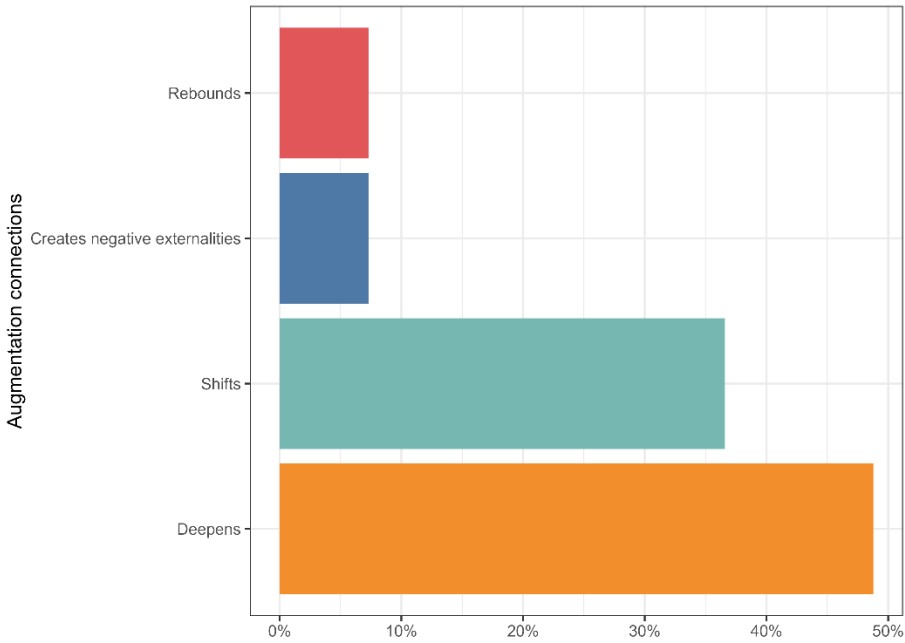

**Figure 10: Proportion of vulnerability augmentation connections in the enhanced Impact Chain**

### 4.2.3. Derived impacts

When augmented, certain vulnerabilities act like impacts and reinforce the impact that increased the vulnerability in the first place. Such augmented vulnerabilities with double status were doubled in the enhanced Impact Chain and called "derived impacts". There are 15 cases where the augmentation of a vulnerability created a derived impact, but only 9 derived impacts

in the Impact Chain, as shown in Appendix B. This means that several vulnerabilities were transformed into derived impacts more than once, by different impacts that generated the augmentation of the vulnerability that was doubled as a derived impact: low-performance medical system (3 times), insufficient medical personnel (3 times), insufficient COVID-19 testing capacity (2 times), uncooperative population (2 times). On the other hand, the vulnerability called households at short distance from the river, insufficient/ineffective hard engineering infrastructure/measures, improper mapping and visualisation of affected areas,

lack of equipment for first responders (including protective gear), and work overload on first responders were transformed into derived impacts only once.

Of the vulnerabilities that also act as derived impacts, 44.44% pertain to floods, 33.33% to the pandemic, and 22.22% to both hazards. More than half (60%) of the derived impacts are associated with "deepens" connections, suggesting that the augmentation of the vulnerabilities and their subsequent reinforcement as derived impacts are mostly related to the same

hazard. All the identified derived impacts are detailed in Appendix B, with the focus in this section limited to the most significant ones.



The augmentation of the low-performance medical system was caused by the effects of the pandemic on other diseases, the economic loss caused by both floods and the pandemic, and the economic challenges brought by the pandemic (Appendix B). In the first instance, the COVID-19 pandemic delayed the provision of treatment for certain diseases (Cucu et al. 2021, Dionisie et al. 2022, Barbos et al. 2023), or accelerated the progression of diseases like kidney pathology (Trifanescu et al. 2022, Mureșan et al. 2022, Tudora et al. 2023). These circumstances exerted additional strain on the already suboptimal medical system, contributing to the exacerbation of other health issues. In addition, the economic loss and the pandemic-related economic challenges have the potential to perpetuate the underfunding of the medical system, with negative effects on its performance. In return, the underperforming medical system is a cause of both economic loss (due to treatment delays and shortages of medical and human resources) and economic challenges stemming from its coping ineffectiveness.

The augmentation of insufficient medical personnel was linked to impacts like human casualties, the effects of the pandemic on other diseases, and increased stress or anxiety (Appendix B). The victims of COVID-19 included healthcare staff that became infected with the virus while attending to COVID-19 patients, which deepened the shortage of personnel and subsequently significantly altered their capacity to provide life-saving healthcare to the thousands of patients in need, therefore increasing the human death toll. Similarly, the surge in workload for medical personnel, resulting from aggravated diseases against the COVID-19 pandemic, limited the availability of healthcare staff dedicated to tending to COVID-19 patients. Consequently, the insufficient number of healthcare personnel negatively affected the development of certain diseases, as timely and appropriate treatment was not administered. Lastly, the increased stress/anxiety temporarily affected the mental health and wellbeing of the medical staff, necessitating temporary breaks in their duties. The temporary unavailability of their colleagues heightened the stress/anxiety levels among the remaining healthcare professionals, as well as among the general public, who was aware of the scarcity of medical human resources during critical times.

The insufficient COVID-19 testing capacity was augmented by the road transportation impairment resulting from floods and also by the disrupted ambulance service. During and immediately after floods, people were precluded from reaching COVID-19 testing centres, and ambulances were prevented from reaching the people who requested to be tested at home. Both of these obstructions limited the testing capacity. In return, the limited testing capacity at local scale forced the people to undertake road journeys to the available testing centres located in other settlements, sometimes at great distances, resulting in traffic jams in numerous places and occasions.

In the analysed multi-hazard case study, the population became even more uncooperative because of the diminished trust in authorities and the increased stress/anxiety associated with both floods and the pandemic. The lessened credibility of authorities can be traced back to the faulty pandemic management and the lockdown imposed in March-May 2020 (Džakula et al. 2022), and also to the economic problems resulting from both hazards. This increased reluctance to collaborate with first responders and authorities also undermined trust in authorities, establishing a positive feedback loop. Amid flood-related interventions, the escalation of stress or anxiety levels can make people fearful and less willing to collaborate with first responders, hindering rescue or evacuation operations. Conversely, this reluctant attitude of the population and the associated difficulties can increase the stress/anxiety of first responders on duty.





### 4.2.4. Ranking of augmented vulnerabilities

Table 5 illustrates the ranking of vulnerabilities in terms of augmentation and the values used to compute it. The top 3 augmented vulnerabilities were: the uncooperative population, low-performance medical system, and flood management not adapted to the COVID-19 context. The first and third are multi-hazard vulnerabilities that correspond to both floods and the pandemic, while the low-performance medical system is specific to the pandemic. In terms of type, the most augmented vulnerability relates to coping deficiencies, while the next two pertain either to medical management or emergency management failures. As for the scale of manifestation, the uncooperative population is a local-level vulnerability, while the other two manifest at a broader, national scale.

The above said vulnerabilities were followed by other management-related vulnerabilities, like the insufficient medical personnel, the lack of equipment for first responders, the shallow implementation of preventive measures, the insufficient COVID-19 testing capacity or ICU capacity, and the work overload on first responders, most of them relating to the COVID-19 pandemic (Table 5). The least augmented vulnerabilities are specific to the flood hazard (i.e., defective coordination of first responders from multiple counties, deforestation, households at short distance from the river, etc.).

When looking at the augmentation produced by impacts, the ranking resembles the final one (Table 5), which is expected due to the 70% weight of the impacts-vulnerability augmentation connections. The difference is that flood management not adapted to the COVID-19 context, the insufficient medical personnel and lack of equipment for first responders were augmented by impacts to equal extents, which also holds true for the next four vulnerabilities. On the other hand, the vulnerabilities that were most augmented by adaptation options (to equal extents) were the shallow implementation of preventive measures and the absence of preparedness at individual level. In the ranking of augmented vulnerabilities by adaptation options, these were followed by the uncooperative population and flood management not adapted to the COVID-19 context which both occupy the third place (Table 5). All of the other vulnerabilities were not augmented by adaptation options.

The vulnerabilities at the bottom of Table 5 were not augmented; however, they bear the potential for escalation due to the fact that they were not addressed by any adaptation options: assets at short distance from river, depleted capacity due to seasonal patterns of hazards, development of infrastructure or inhabited areas in flood-prone areas, improper governance structure for effective flood management, ineffective sewage system, and low quality construction materials. Except for the depleted capacity due to seasonal patterns of hazards, all of these are specific to floods.

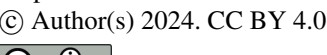

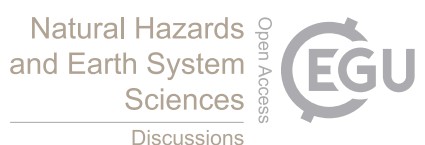
**Table 5. Ranking of vulnerabilities based on their augmentation. I – Impact, V – Vulnerability, Ao – Adaptation option**

| Vulnerability | No. of I-V augmentation connections | No. of Ao-V augmentation connections | Z-score I-V | Z-score Ao-V | Rank Z-score I-V | Rank Z-score Ao-V | Weighted rank | Final rank |
|---|---|---|---|---|---|---|---|---|
| Uncooperative population | 5 | 1 | 2.640 | 1.310 | 1 | 3 | 0.8 | 1 |
| Low-performance medical system | 4 | 0 | 1.918 | -0.393 | 2 | 5 | 1.45 | 2 |
| Flood management not adapted to the COVID-19 context | 3 | 1 | 1.195 | 1.310 | 3 | 3 | 1.5 | 3 |
| Insufficient medical personnel | 3 | 0 | 1.195 | -0.393 | 3 | 5 | 1.8 | 4 |
| Lack of equipment for first responders (including protective gear) | 3 | 0 | 1.195 | -0.393 | 3 | 5 | 1.8 | 4 |
| Shallow implementation of preventive measures | 2 | 2 | 0.472 | 3.014 | 6 | 1 | 2.25 | 6 |
| Insufficient COVID-19 testing capacity | 2 | 0 | 0.472 | -0.393 | 6 | 5 | 2.85 | 7 |
| Insufficient ICU capacity (e.g., no. of beds, ventilators, O2 supply) | 2 | 0 | 0.472 | -0.393 | 6 | 5 | 2.85 | 7 |
| Work overload on first responders | 2 | 0 | 0.472 | -0.393 | 6 | 5 | 2.85 | 7 |
| Improper mapping and visualisation of affected areas | 2 | 0 | 0.472 | -0.393 | 6 | 5 | 2.85 | 7 |
| Defective coordination of first responders from multiple counties | 1 | 0 | -0.250 | -0.393 | 11 | 5 | 4.6 | 11 |
| Deforestation | 1 | 0 | -0.250 | -0.393 | 11 | 5 | 4.6 | 11 |
| Households at short distance from the river | 1 | 0 | -0.250 | -0.393 | 11 | 5 | 4.6 | 11 |
| Insufficient/ineffective hard engineering infrastructure/measures | 1 | 0 | -0.250 | -0.393 | 11 | 5 | 4.6 | 11 |
| Long shifts of first responders | 1 | 0 | -0.250 | -0.393 | 11 | 5 | 4.6 | 11 |
| Poverty, especially in uneducated/roma/migrant population | 1 | 0 | -0.250 | -0.393 | 11 | 5 | 4.6 | 11 |
| Significant psychological tension of first responders | 1 | 0 | -0.250 | -0.393 | 11 | 5 | 4.6 | 11 |
| Absence of preparedness at individual level | 0 | 2 | -0.973 | 3.014 | 18 | 1 | 6.45 | 18 |
| Assets at short distance from river | 0 | 0 | -0.973 | -0.393 | 18 | 5 | 7.05 | 19 |
| Depleted capacity due to seasonal patterns of hazards | 0 | 0 | -0.973 | -0.393 | 18 | 5 | 7.05 | 19 |
| Development of infrastructure in flood prone areas | 0 | 0 | -0.973 | -0.393 | 18 | 5 | 7.05 | 19 |
| Development of inhabited areas in flood prone areas | 0 | 0 | -0.973 | -0.393 | 18 | 5 | 7.05 | 19 |
| Improper governance structure for effective flood management | 0 | 0 | -0.973 | -0.393 | 18 | 5 | 7.05 | 19 |
| Ineffective institutional communication | 0 | 0 | -0.973 | -0.393 | 18 | 5 | 7.05 | 19 |




| | | | | | | | | |
|---|---|---|---|---|---|---|---|---|
| Ineffective sewage system | 0 | 0 | -0.973 | -0.393 | 18 | 5 | 7.05 | 19 |
| Low quality construction materials | 0 | 0 | -0.973 | -0.393 | 18 | 5 | 7.05 | 19 |
| **Average** | 1.346 | 0.231 | | | | | | |
| **Standard deviation** | 1.384 | 0.587 | | | | | | |





## 5. Discussion

The current study stands at the forefront of research, bringing into the spotlight the increase in vulnerability within the unprecedented co-occurrence of the COVID-19 pandemic and the multiple flood events that affected Romania in 2020-2021. The configuration of the Impact Chain shows a convoluted multi-hazard, where certain hazard impacts and adaptation options

have an augmentation effect on underlying vulnerabilities. In return, some of the augmented vulnerabilities also act as derived impacts that reinforce the very impacts that increased vulnerability in the first place. In this sense, both hazards and what we do to mitigate them can be considered indirect generators of changes in vulnerability, with deep implications for how we approach multi-risk management.

In the presented case study, the enhanced Impact Chain shows that vulnerability is expected to increase based on the

augmentation in different forms conveyed by the new links – as 69.23% of vulnerabilities were augmented either by impacts, or backfiring adaptation options, and also based on the limited range of adaptation options that address vulnerabilities – as the same percentage corresponds to the vulnerabilities that were not addressed by any adaptation options. This means that 1) the unforeseen implications of impacts that act as vulnerability enhancers, 2) wrongful action intended to mitigate vulnerability and/or impacts, and 3) inaction can set the premises for increased vulnerability levels that will render multi-risk management

more difficult (Figure 11).

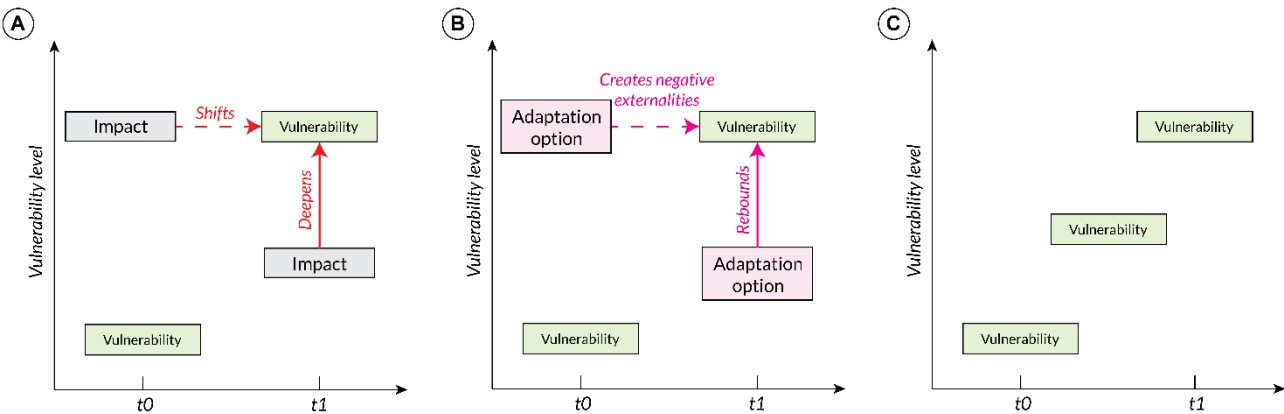

**Figure 11: Trajectories of rising vulnerability: A – Augmentation of vulnerability resulting from hazard impacts, B – Augmentation of vulnerability resulting from misfiring adaptation options, C – Perpetuation of vulnerability due to inaction. t0 – Present moment, t1 – Future moment**

## 5.1. Trajectories of rising vulnerability

The first trajectory refers to the impacts of the flood events and the pandemic (Figure 11A). These mainly reinforce deeply rooted vulnerabilities, like the reluctance of the population to collaborate with first responders and/or authorities (Fekete et al. 2023), or the low-performance of the Romanian medical system that is largely reported in the literature (OECD 2021, Lupu





and Tiganasu 2022, Popescu et al. 2022). The top 3 most impact-augmented vulnerabilities also includes the deficiency in
aligning flood management with pandemic conditions (Table 5), which was associated with local increases in the COVID-19
new cases (Albulescu 2023) and is expected to cause further issues in similar future multi-hazard scenarios unless amended.
The top of the ranking of impact-augmented vulnerabilities also includes other medical or emergency management
vulnerabilities (i.e., insufficient medical personnel, lack of equipment for first responders). All of the above said vulnerabilities
were addressed by various adaptation options, but most of them produced short-term effects and were not part of larger
vulnerability reduction schemes. Several examples are the on-the-spot clever solutions brought up by first responders to engage
with the uncooperative population, the hiring of additional medical staff and volunteers during the pandemic, and the support
received by Romania from other countries in terms of medical resources.

The second line along which the augmentation vulnerability propagates is established between the adaptation options that
misfire and end up increasing vulnerabilities (Figure 11B). The most augmented vulnerabilities in this regard concern the
preparedness phase of DRM: the shallow implementation of preventive measures against the COVID-19 pandemic and the
absence of preparedness at individual level when confronted with floods (Table 5). The low level of preparedness was
associated with an external locus of control (Armaș 2008, Armaș et al. 2015, Albulescu et al. 2021), and also reported by first
responders who performed interventions during the floods of 2020-2021 (Fekete et al. 2023). This analysis unravels the
possibility that these coping capacity-related vulnerabilities can evolve into (vulnerability) drivers. Against this background,
a major gap emerges between the efforts undertaken by first responders in the response phase, and the lack of interest on the
part of citizens, who take no or little action to prepare to withstand floods or to prevent the spread of the SARS-CoV-2 virus
during the preparedness phase.

Another thing to consider is that the top 3 impact-augmented vulnerabilities are the same as the ones that rank vulnerabilities
based on the combined augmentation effects of impacts and adaptation options. However, the vulnerabilities that were
augmented by both impacts and adaptation options (to different extents) are the uncooperative population, the lack of
adaptation of flood management to pandemic conditions, and the shallow implementation of preventive measures against the
pandemic (Table 5). In future multi-risk management plans, special emphasis should be placed on addressing these
vulnerabilities, particularly given that the first two are related to both hazards.

The third trajectory of increasing vulnerability is through inaction (Figure 11C), standing out since the number of
vulnerabilities (26) is two times larger than the ones of adaptation options (13), and only about a third of the vulnerabilities
were targeted by adaptation options. When looking at the entire enhanced Impact Chain, a striking imbalance is highlighted:
most flood-related mitigation efforts focused on impacts rather than vulnerabilities, while pandemic-related adaptation options
primarily addressed vulnerabilities rather than pandemic impacts. The only flood-related vulnerability addressed by adaptation
options is the improper mapping and visualisation of affected areas. This means that human communities might be equally or
more vulnerable to floods in the future. What is more, even the adaptation options that mitigated the flood impacts mostly
provide short-term solutions (e.g., the heightening of river banks with sand banks to prevent or limit the flooding of houses or
households) or have negative unforeseen effects (e.g., the RO-Alert SMS messages or the hydrological warnings that can





reduce the motivation of the people who are not located in an area affected by a particular flood event to prepare for future floods or to undertake COVID-19 prevention measures, as described in Appendix B).

On the contrary, many of the key pandemic vulnerabilities were tackled by adaptation options (e.g., low-performance medical system, insufficient medical personnel, insufficient ICU capacity), and the same can be stated for multi-hazard vulnerabilities (e.g., flood management not adapted to the COVID-19 context, ineffective institutional communication, lack of equipment for first responders, uncooperative population) (Appendix A). Nevertheless, the brighter perspective described here is overshadowed by the fact that the very same vulnerabilities (except the ineffective institutional communication) were

augmented by hazard impacts and/or adaptation options.

This approach leaves deeply engrained vulnerabilities to floods unaltered (e.g., the location of households and/or assets at short distance from the river, the improper governance structure for effective flood management, the shallow implementation of the absence of individual flood preparedness), but ready to resurface during future hazardous events. In other words, the implemented adaptation options belong to the response and/or recovery phase of the DRM, and no initiatives have been

undertaken in the preparedness phase. What is worse, as argued above, is that certain adaptation options augment the two prominent vulnerabilities specific to the preparedness phase (e.g., the shallow implementation of COVID-19 preventive measures, and the absence of individual flood preparedness measures). The reactive approach is typical to developing societies, or to the early, one-dimensional flood management approach (Scott et al. 2013), being complemented by an external locus of control of the population (Armaș 2008, Armaș et al. 2015). Sound risk mitigation requires factoring in preparedness for future

hazards into the recovery process (Johnson and Jensen 2023), all with a high degree of flexibility (White and Haughton 2017), but such efforts were absent in the presented case study. Therefore, the unbalanced DRM-phase distribution of the adaptation options holds prominent implications for the dynamics of vulnerabilities in the sense that it allows them to perpetuate and further contribute to future hazard impacts.

Another aspect to ponder is that the depleted capacity due to seasonal patterns of hazards, although not augmented, was not

addressed by any adaptation options. Both floods and pandemic waves follow seasonal patterns, allowing human communities to prepare for their impacts (to some extent) by following a predictive but tight timeline. Considering the unaddressed vulnerabilities together with the short-sighted nature of the adaptation options, human communities affected by the COVID-19 pandemic did not fully recover until the occurrence of floods, or until the next pandemic wave, or perhaps not even from one flood to the next. In this context, it can be expected that the overall vulnerability level will increase, since the recovery

process is not only slow (de Ruiter and van Loon) but also fragmented.

The short time intervals between pandemic waves, which unfold during the cold months of the year (Figure 3), and the clusters of flood events at the end of spring and beginning of summer require expedite mitigation efforts and updated multi-risk management plans that account for the particularities of the co-occurrent hazards. This holds particular importance since the most prominent adaptation option is the great capacity of first responders to develop creative solutions in crisis and cope with

new challenges. This is the only adaptation option that mitigates multiple top-level, augmented vulnerabilities that pertain to both hazards: the uncooperative population, flood management not adapted to the COVID-19 context, lack of equipment for





first responders including protective gear, ineffective institutional communication. The umbrella adaptation option covers a large spectrum of mitigation actions thought about and implemented by first responders on the spot, to cover for the lack of specific protocols. This means that there are no adaptation options that account for the challenges imposed by the two
independent but co-occurrent hazards, highlighting a lack of vision of the current risk management plans applied in Romania.

### 5.2. Limitations and constraints

Pursuing scientific rigour and transparency, the limitations of the study have to be acknowledged too. The case study aimed for a comprehensive analysis of the multi-hazard of interest, drawing on various data and information sources. However, this is only as comprehensive as possible, given the fact that there are no official sources that detail the impact of flood events.
Also, the exact quantification of the impacts is constrained by the lack of official data. Along the same line, the absence of information on the COVID-19 preventive measures implemented during flood evacuation procedures and inside emergency shelters raises uncertainties that are integrated into the Impact Chain. Another shortcoming concerns the limited time range that does not cover the entire pandemic period but only its first two years. It should be mentioned that 2022 was a dry year in Romania (Iuga 2022), implying that flood occurrences were scarce. In addition, Albulescu (2023) reports no flood events that
required the evacuation of the population in the first 8 months of 2022 (including the flood season in Romania). A fourth limitation regards the tangled configuration of the Impact Chains, which does not allow for a figure-based visualisation in the paper. Nevertheless, the visualisation available via the Kumu links provided in Table 3 holds the advantage of interactive manipulation of connections and elements, as well as access to the descriptions, source types, references, maps, and images embedded in the Impact Chains. Nevertheless, a comprehensive understanding of the paper is facilitated by engaging with the
online platform.

### 5.3. Contribution and novelty

This study marks a pioneering research work, breaking new ground by addressing the augmentation of vulnerability that arises from the impact of co-occurrent hazards. Interest in vulnerability dynamics has surfaced since 2020, and discussions have remained at a theoretical level (de Ruiter and Van Loon 2022), with no case study up to date. Moreover, few studies
investigated the interactions between flood hazard and the COVID-19 pandemic (Simonovic et al. 2020, Patwary and Rodriguez-Morales 2021, Pramanik et al. 2021, Turay 2022). This paper addresses a double research gap, aiming to advance our understanding of both vulnerability variations against a multi-hazard background and of compounded impacts of the two hazards of interest.

The methodological framework proposed to reach this goal carries multiple elements of novelty, as it enhances the Impact
Chain to account for the fluctuations in vulnerability by establishing new elements and connection types and taking an in-depth look at the double status of certain augmented vulnerabilities (i.e., those that also act as derived impacts). The enhanced Impact Chain is a readily available operational tool suitable for replication across various multi-hazard, timeframes, scales, and geographic settings. This improved version of the chain can extend the list of methods for vulnerability dynamics





modelling put together by de Ruiter and Van Loon (2022), also emerging as a solution to the issue raised by Tilloy et al.

(2019): "We believe there is a need to not only study case studies inclusive of multi-hazard interrelationships but to generalise to more inclusive frameworks that are applicable to a broad range of hazards and locations." The dual functionality highlights the capability of the methodological framework to account for both changes in vulnerability and the intricacies of multi-hazard impacts, working as a documentation, diagnosis, and prediction tool.

It should be noted that the present analysis on the augmentation of vulnerability against a multi-hazard background is an initial

research work. Prospective avenues for research include the development of a model of systemic vulnerability in a multi-hazard context, which will be tested on multiple Impact Chains, including the enhanced one discussed in this study, to further validate its effectiveness and applicability.

## 6. Conclusions

Since the start of the decade, the co-occurrence of natural hazards amid the COVID-19 pandemic put us in front of unparalleled

challenges that demanded a new way of approaching multi-hazard management and adaptability to both public health crisis and the impacts of various natural hazards. This increase in multi-hazard frequency thought us valuable lessons that we still have to untangle in the years to come in order to reduce our vulnerability in face of future similar multi-hazard events.

Here, we posit that particular emphasis should be placed on understanding the dynamics of vulnerability within a multi-hazard context, and that we still have to develop the tools for analysis focusing on the fluctuations of vulnerability across hazards,

time, and space. To this end, we enhanced the Impact Chain regarding the multi-hazard of the floods and COVID-19 pandemic that affected Romania in 2020-2021, transforming it from a documentation tool to a diagnosis and prediction one. The main enhancements are: the introduction of new types of elements (i.e., augmented vulnerabilities, derived impacts), new types of connections between the impacts/adaptation options and vulnerabilities, and the ranking of vulnerabilities based on their augmentation.

The key findings of the paper can be summarised as follows:

- In a multi-hazard context, vulnerability can be augmented by both impacts and adaptation options in ways that can be captured by an Impact Chain, but it can also perpetuate over time due to inaction to address it (Figure 11).

- Certain augmented vulnerabilities can also be considered impacts (here called "derived impacts") that sharpen the impact that initiated the augmentation of that vulnerability in the first place.

- In the case study of the floods and the COVID-19 pandemic in Romania (2020-2021), vulnerability is augmented mostly by hazard impacts and, to a lesser extent, by adaptation options.

- Vulnerability is expected to increase due to inaction (as certain vulnerabilities were not addressed by adaptation options), through the unforeseen implication of hazard impacts, or through the misfiring of adaptation options.





• The main multi-hazard impacts relate to the damages to houses and infrastructure (particularly roads and lifelines), the (self-) evacuation/displacement of people, the potential increase of COVID-19 cases at local scale, and the flooding of hospitals.

• The most augmented vulnerabilities (by both impacts and adaptation options) in the proposed Impact Chain are: uncooperative population, low-performance medical system, flood management not adapted to the COVID-19 160 context.

• The most augmented vulnerabilities by adaptation options alone (i.e., shallow implementation of preventive measures and absence of preparedness at individual level) show that adaptation options undermine preparedness to both floods and the pandemic.

These results reinforce the idea that old ways will not solve new or reinforced problems and that a proper understanding of all 165 components of multi-risk – and especially of those that can be mitigated (i.e., impacts and vulnerabilities), is the key to improving multi-risk management. The Impact Chain brings to light the shallow approach of multi-hazard management in Romania, which fails to cover all three DRM phases (i.e., preparedness, response, recovery) or to account for the co-occurrence of multiple hazards and to raise to the challenges in the last years. Such situations motivate the need for improved "multi-hazard approach and inclusive risk-informed decision-making" mentioned in the Sendai Framework for Disaster Risk 170 Reduction 2015-2030. Although such goals were set before the COVID-19 pandemic, their achievement is still an ongoing process, the progress of which hinges on our understanding of the dynamics of multi-hazard vulnerability.

*Data availability.* The data can be provided by the authors upon reasonable request.

*Author contributions.* ACA and IA contributed equally to this study during all stages: conceptualisation, Impact Chain 175 construction, ranking, calibration, writing the first draft, editing, and validation.

*Competing interests.* The authors declare no conflict of interest.

*Acknowledgements.* This publication was developed within the EU PARATUS project. It has received funding from the European Union's Horizon Europe research and innovation programme under grant agreement No. 101073954 and from UEFISCDI PN-IV-P8-8.1-PRE-HE-ORG-2023-0120. We acknowledge the support of PARATUS Deliverable 1.1 2023.

*Financial support.* This was supported by the EU PARATUS project, European Union's Horizon Europe research and innovation programme under grant agreement No. 101073954, and by and from UEFISCDI PN-IV-P8-8.1-PRE-HE-ORG-2023-0120.

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



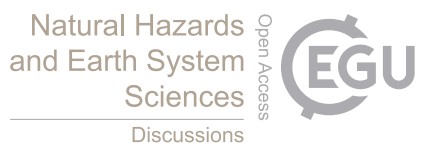
**Appendix A. Vulnerabilities grouped by hazard, type, and scale**

| Hazard | Vulnerability | Type of vulnerability | Scale | Mitigated | Augmented |
|---|---|---|---|---|---|
| COVID-19 pandemic | Insufficient COVID-19 testing capacity | Vulnerability related to medical management | National | No | Yes |
| COVID-19 pandemic | Insufficient ICU capacity (e.g., no. of beds, ventilators, O2 supply) | Vulnerability related to medical management | Local | Yes | Yes |
| COVID-19 pandemic | Insufficient medical personnel | Vulnerability related to medical management | Local | Yes | Yes |
| COVID-19 pandemic | Low-performance medical system | Vulnerability related to medical management | National | Yes | Yes |
| COVID-19 pandemic | Shallow implementation of preventive measures | Vulnerability related to medical management | Local | No | Yes |
| Floods | Absence of preparedness at individual level | Vulnerability related to coping capacity | Individual | No | Yes |
| Floods | Assets at short distance from river | Vulnerability related to territorial planning | Local | No | No |
| Floods | Defective coordination of first responders from multiple counties | Vulnerability related to emergency management | Regional | No | Yes |
| Floods | Deforestation | Vulnerability related to territorial planning | Local | No | Yes |
| Floods | Development of infrastructure in flood prone areas | Vulnerability related to territorial planning | Local | No | No |
| Floods | Development of inhabited areas in flood prone areas | Vulnerability related to territorial planning | Local | No | No |
| Floods | Households at short distance from the river | Vulnerability related to territorial planning | Local | No | Yes |
| Floods | Improper governance structure for effective flood management | Vulnerability related to emergency management | National | No | No |
| Floods | Improper mapping and visualisation of affected areas | Vulnerability related to emergency management | Local | Yes | Yes |
| Floods | Ineffective sewage system | Vulnerability related to infrastructure | Local | No | No |
| Floods | Insufficient/Ineffective hard engineering infrastructure/measures | Vulnerability related to infrastructure | National | No | Yes |
| Floods | Long shifts of first responders | Vulnerability related to emergency management | Local | No | Yes |
| Floods | Low quality construction materials | Vulnerability related to infrastructure | Local | No | No |

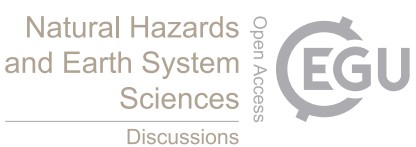
*Appendix B. Details on the new connection types and derived impacts included in the Enhanced Impact Chain. The \* marks the cases where the impact in the first column or the vulnerability in the second column relate both to floods and the COVID-19 pandemic. In such cases, the type of augmentation can be both deepening and shifting vulnerability, but the choice was based on the explanation given in the fourth column.*

| Impact/ Adaptation option | Augmented vulnerability | Type of augmentation (New connection) | Vulnerability relation | Scale | | | Explanation of augmentation | Augmented vulnerability turned into derived impact |
|---|---|---|---|---|---|---|---|---|
| Floods | Significant psychological tension of first responders | | Vulnerability related to emergency management | Local | No | Yes | | |
| Floods | Work overload on first responders | | Vulnerability related to emergency management | Local | No | Yes | | |
| Floods, pandemic COVID-19 | Depleted capacity due to seasonal patterns of hazards | | Vulnerability related to coping capacity | National | No | No | | |
| Floods, pandemic COVID-19 | Flood management not adapted to the COVID-19 context | | Vulnerability related to emergency management | National | Yes | Yes | | |
| Floods, pandemic COVID-19 | Ineffective institutional communication | | Vulnerability related to emergency management | Local | Yes | No | | |
| Floods, pandemic COVID-19 | Lack of equipment for first responders (including protective gear) | | Vulnerability related to emergency management | Local | Yes | Yes | | |
| Floods, pandemic COVID-19 | Poverty, especially in population | | Vulnerability related to coping capacity | Local | No | Yes | | |
| Floods, pandemic COVID-19 | Uneducated/roma/migrant population | | Vulnerability related to coping capacity | Local | Yes | Yes | | |
| Floods, pandemic COVID-19 | Uncooperative population | | Vulnerability related to coping capacity | Local | Yes | Yes | | |
| Water contamination | Households at short distance from the river | Deepens vulnerability | | | | | Floods can contaminate the water of rivers, which fosters water-borne diseases. Human communities located close to rivers are especially exposed to such contamination, which makes them more vulnerable (to floods and diseases). | Households at a short distance from the river, under specific environmental and river valley morphology conditions, will increase water contamination issues downstream. |
| Railway transportation impairment | Shallow implementation of preventive measures | Shifts vulnerability | | | | | Flood-determined railway transportation impairment can increase the vulnerability of travellers to COVID-19 by causing unnecessary crowding of trains or prolonged exposure due to delays. This is particularly relevant since few preventive measures were implemented to limit the spread of the SARS-CoV2 virus during train travel. | |
| Lockdown | Deforestation | Shifts vulnerability | | | | | The lockdown imposed in March-May 2020 favoured the illegal cutting of the forest, especially in mountainous, isolated areas. As a protective measure, forest authorities decided to guard the forests. | |
| Disrupted ambulance service | Insufficient COVID-19 testing capacity | Shifts vulnerability | | | | | In the early pandemic months, the COVID-19 testing of the population was done by calling the ambulance and requesting to be tested. During or after floods, ambulances could not reach the potential COVID-19 patients, which deepened the limitation of the testing capacity. | The vulnerability of insufficient COVID-19 testing capacity also acts as a derived impact, since this limitation in testing caused disruption in the functioning of the ambulance service. During the pandemic, ambulances worked at full capacity, especially for testing |


| Event | Vulnerability driver | Effect | Description |
|---|---|---|---|
| Interventions to remove flood water | Flood management not adapted to the COVID-19 context | Rebounds vulnerability | The emergency management personnel in charge of removing the flood water and cleaning after a flood event are exposed to COVID-19 during these operations that minimise social distancing. This prolonged contact with each other, the population, and contaminated water increases the vulnerability that stems from the absence of adaptation of flood management protocols to the pandemic conditions. |
| Potential increase in the COVID-19 new cases | Insufficient ICU capacity (e.g., no. of beds, ventilators, O2 supply) | Deepens vulnerability | The potential increase in the COVID-19 positive cases augments the pressure on the ICUs that already function at full capacity. |
| Effects on other diseases | Low-performance medical system | Deepens vulnerability | The COVID-19 diseases delayed the provision of health care to non-COVID patients (with the effect of aggravating their pre-existing disease), therefore reducing the performance of the medical system and also increasing the vulnerability of non-COVID patients. Under additional pandemic pressure, the low-performance medical system will have a derived impact with multiple adverse effects on other diseases patients suffer from. |
| Human casualties | Insufficient medical personnel | Deepens vulnerability* | The death toll of COVID-19 among medical personnel reduced the number of health care professionals available to carry on the fight against the pandemic. The shortage of medical personnel represents a derived impact that increases the number of human casualties, since many COVID-19 and non-COVID-19 patients in need of health care could have been saved if they would have benefited from medical attention. |
| RO-Alert SMS messages | Shallow implementation of preventive measures | Rebounds vulnerability | The RO-Alert SMS messages issued as part of the emergency-related communication to the population in the context of floods may have caused panic, increasing the chances of abandoning the protective behaviour against the COVID-19 infection. |
| Flooded public buildings (including 1 hospital) | Low-performance medical system | Shifts vulnerability | The flooding of buildings that host hospitals appears as a supplementary problem that contributes to the low performance of the medical system, diverting financial resources from other pressing issues. |
| Cut off supply of electricity/gas/water | Insufficient ICU capacity (e.g., no. of beds, ventilators, O2 supply) | Shifts vulnerability | The frequent blackouts of electricity/gas/water that occur during or immediately after flood events can greatly impact the functionality of ICUs, limiting their capacity to provide health care. |
| Effects on other diseases | Insufficient medical personnel | Deepens vulnerability | The medical personnel have to face increased workloads because the COVID-19 infection aggravates the pre-existing diseases of patients. These complex situations reduce the personnel available to tend to COVID-19 patients in certain medical units. Insufficient medical personnel is not only a vulnerability but also a derived impact. The shortage of doctors and nurses can also contribute to the progression of certain diseases that were already aggravated by the infection with SARS-CoV2. |
| Flooded/Damaged households or houses | Flood management not adapted to the COVID-19 context | Shifts vulnerability* | The flooding of houses or households determines the evacuation of the population, a procedure that is not adapted to the new pandemic conditions. During evacuation operations, people get in close contact with each other, favouring the spread of the COVID-19 infection. Also, the evacuees that are accommodated in temporary shelters are exposed to the spread of the virus. |
| Economic loss | Low-performance medical system | Shifts vulnerability* | The economic loss resulting from the flood events or the pandemic perpetuates the low performance of the medical system because of the implicit, "chronical" lack of financial support. At the same time, the low performance of the medical system determines economic loss, in direct relation to treatment delays, a shortage of medical and human resources, etc. |
| Displaced/(Self-) Evacuated people | Flood management not adapted to the COVID-19 context | Shifts vulnerability* | The evacuation procedures performed before, during, or after floods increase the vulnerability of the evacuees and/or the emergency management staff, who get in close contact with each other and are exposed to the spread of the |

or other COVID-19-related emergencies, to the expense of the non-COVID patients that also requested health care.

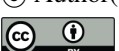


| Impact | Vulnerability factor | Effect | SARS-CoV-2 virus both during transportation and inside temporary emergency shelters. | |
|---|---|---|---|---|
| Road transportation impairment | Insufficient COVID-19 testing capacity | Shifts vulnerability | The flood-induced damages to the road infrastructure, together with the subsequent road transportation impairment, limited the capacity of COVID-19 testing, since people were unable to reach testing centres and ambulances were unable to reach the people requesting to be tested at home. | The insufficient COVID-19 testing capacity also caused road transportation impairment, as many people were unable to get tested or vaccinated in the settlement of residence and close to undertake road journeys on different distances (considerable, in some cases) to available testing/vaccination centres located in other settlements. This COVID-19-related transportation boost caused traffic jams in numerous places and on occasions. |
| Increased stress/anxiety | Insufficient medical personnel | Deepens vulnerability* | The increased stress/anxiety during the pandemic waves severely affected the mental health and wellbeing of the medical personnel. In certain cases, the doctors or nurses became unable to perform their medical duties, even for short periods of time, which deepened the shortage of medical personnel at different times. | The insufficient medical personnel also represents a derived impact, since it is an additional cause of stress/anxiety for both the existing medical staff and the patients, or the general population. |
| Vaccination campaign against the SAR-CoV-2 virus | Shallow implementation of preventive measures | Creates negative externalities | The vaccination campaign had the unwanted effect of diluting the interest in implementing early COVID-19 prevention measures (e.g., the wearing of masks, social distancing). In certain instances, even unvaccinated people can lower their guard in self-protection, assuming that being surrounded by vaccinated individuals prevents them from contracting infections. | |
| Cut off supply of electricity/gas/water | Shallow implementation of preventive measures | Shifts vulnerability | The cut off of energy/gas/water may cause people to gather in neighbours' or relatives' houses, which reduces social distancing and increases the chances of COVID-19 infection. | |
| Economic loss | Insufficient/ineffective hard engineering infrastructure/measures | Shifts vulnerability* | The economic loss sets the premises for underfunding the insufficient/ineffective hard engineering infrastructure/measures. | Insufficient/ineffective hard engineering measures are a derived impact, since they provoke an increase in economic losses in the case of exposed and vulnerable communities. |
| Economic challenges | Low-performance medical system | Deepens vulnerability | The economic challenges resulting from the COVID-19 pandemic may divert attention and financial resources from improving the medical system, accentuating its low performance. | A low-performance medical system will be a derived impact raising new economic challenges due to its ineffectiveness in coping. |
| Heighten river banks with sand bags | Absence of preparedness at individual level | Rebounds vulnerability | The implementation of last-minute, on-the-spot solutions like the heightening of river banks with sand bags offers a false impression of security, reducing the interest of people in preparedness at household level. | |
| Flooded basement hospital | Flood management not adapted to the COVID-19 context | Shifts vulnerability | The flooding of hospital buildings increases the vulnerability of emergency management staff who have to remove the water, perhaps entering contaminated areas and getting in contact with medical personnel or patients who are infected with COVID-19. The patients infected with the SARS-CoV-2 virus can be moved into other wards/rooms/buildings, which become overcrowded in the aftermath of a flood event that affects the hospital buildings. Both the gathering and the transportation of the patients increase their exposure to COVID infection and disrupt their health care routines, all leading to increased vulnerability. | |
| Hydrological warnings | Absence of preparedness at individual level | Creates negative externalities | The people located in areas that were not mentioned in the hydrological warnings for one flood event gained a false feeling of security, which can reduce their interest in implementing flood preparedness measures at individual level. | |

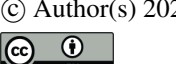


| Cause | Effect | Type | Description | Derived/feedback note |
|---|---|---|---|---|
| Vaccination campaign against the SARS-CoV-2 virus | Uncooperative population | Creates negative externalities | The vaccination campaign against the SARS-CoV-2 virus was accompanied by abundant misinformation that fueled conspiracy theories. This exacerbated the reluctance of people to cooperate with authorities. | The uncooperative population also represents a derived impact, as the lack of support and availability of collaboration with authorities increase stress/anxiety in both parts. |
| Increased stress/anxiety | Uncooperative population | Deepens vulnerability* | The stress/anxiety induced by floods can make people uncooperative in relation to first responders, making rescue or evacuation operations harder to implement. | |
| Deas/Missing domestic animals | Uncooperative population | Deepens vulnerability | The death or disappearance of domestic animals during a flood can make people reluctant to evacuate and wanting to search for their animals. Also, people can put themselves in danger in their endeavour to find and save their missing animals. | |
| Mental health issues (e.g., depression) | Uncooperative population | Shifts vulnerability | One of the notable consequences of mental health issues is a diminished inclination towards collaboration. This escalation of uncooperative behaviour can hinder communication with first responders or medical personnel. | |
| Lockdown | Uncooperative population | Deepens vulnerability | The restrictions imposed by authorities during the lockdown (March-May 2020) meant to tilt the SARS-CoV-2 infection curve negatively affected the freedom of citizens, with the effect of reducing their availability to cooperate and also their trust in authorities. | |
| Diminished trust in authorities | Uncooperative population | Deepens vulnerability* | The eroded trust in authorities determined by the faulty management of the COVID-19 pandemic and fuelled by the resulting economic problems contributed to a diminished spirit of cooperation between the population and first responders. | Conversely, the heightened reluctance to collaborate with first responders and authorities further eroded trust in authorities, creating a positive feedback loop. |
| Road transportation impairment | Work overload on first responders | Deepens vulnerability | The restricted access to certain areas affected by floods because of flood-determined road transportation impairment can increase the work load on first responders, as the possibility of getting more people on the ground where and when needed is limited. This means that the first responders who managed to arrive in the affected areas have to cover more ground without supplementary personnel. | |
| Road transportation impairment | Long shifts of first responders | Deepens vulnerability | Impaired road transportation hinders the arrival of the next shift of first responders in the intervention area, potentially leading to extended shifts for the already deployed responders. | |
| Potential increase in the COVID-19 new cases | Work overload on first responders | Deepens vulnerability* | High local viral loads can increase the risk of SARS-CoV-2 infection for first responders, which means that the work load of the uninfected ones can increase. | Excessive workloads for first responders can contribute to a rise in new COVID-19 cases within their units as their exposure to infection increases |
| Cut off supply of electricity/gas/power | Lack of equipment for first responders (including gear) protecting | Deepens vulnerability | The outages of electricity/gas/power determined by floods can alter the functionality of the equipment used by first responders during flood management interventions. | |
| Road transportation impairment | Lack of equipment for first responders (including gear) protecting | Deepens vulnerability | The obstruction of road transportation caused by floods can prevent first responders from transporting certain equipment in flood-affected areas. | |
| Economic loss | Lack of equipment for first responders (including gear) protecting | Shifts vulnerability* | The economic loss resulting from the flood events or the pandemic can reduce interest in investing in equipment needed for flood-related interventions. | The absence of proper equipment in flood-related interventions can amplify the economic loss caused by floods, as it reduces the capacity to safeguard assets on the ground. |



Natural Hazards
and Earth System
Sciences



Discussions

| | | | | |
|---|---|---|---|---|
| Cut off supply of electricity/gas/power areas | Improper mapping and visualisation of affected areas | Deepens vulnerability | Power outages can affect the functionality of computers and other devices used for the mapping and visualisation of affected areas, hindering flood management, especially on short term. | |
| Economic loss | Improper mapping and visualisation of affected areas | Shifts vulnerability* | The financial setbacks resulting from flood events or the pandemic can divert attention and funds from investing in the technological and human resources involved in mapping and producing visualisations of flood-affected areas. | Inadequate mapping and visualisation of flood-affected areas can increase the economic loss by hindering the acquisition and utilisation of accurate data and information on the ground. |
| Flooded hospital basement | Significant psychological tension of first responders | Deepens vulnerability | The challenges linked to flood mitigation efforts inside buildings housing vulnerable people (e.g., hospitals) can increase the psychological tensions experienced by first responders. | |
| Road transportation impairment | Defective coordination of first responders from multiple counties | Deepens vulnerability | Impaired road transportation obstructs coordination among units of first responders in neighbouring counties, potentially diminishing the effectiveness of flood mitigation actions. | |
| Increased unemployment | Poverty, especially in uneducated/roma/migrant population | Deepens vulnerability | The temporary layoffs prompted by the COVID-19 pandemic exacerbated the poverty of the vulnerable population, most of whom are people with low levels of education, roma, or migrant minorities. | |