# Peer review of "An Impact Chain-based exploration of multi-hazard vulnerability dynamics. The multi-hazard of floods and the COVID-19 pandemic in Romania"

_Natural Hazards and Earth System Sciences, 2024_

## Author Comment (AC1)

**Response to Reviewer 1**

We are grateful to the reviewer for putting in a lot of effort and dedicating time to reading and analysing the manuscript, and for the useful and insightful review comments that definitely improved the outcome of this paper. We committed to diligently addressing the comments, and we hope that we succeeded in satisfying the exigencies and the high academic standards of the reviewer.
 Please find below the point by point responses.

*The reviewers' comments are written in italics*, and our responses in regular font. *We chose blue and italic formatting for citations from the manuscript.*
All of the line numbers refer to the reviewed version of the manuscript.

**R1:** *While the topic and methodology of the paper present some interesting and novel ideas, i feel the manuscript needs a significant amount of work and restructuring before it can be published in NHESS, which i why i am suggesting major revisions to the manuscript. The main challenge is that this paper tried to do too much. It is hard to follow the narrative/ red thread, and i found myself getting lost in the many steps of the methodology, which is rather eclectic in nature. I feel the paper could benefit from a reduction in steps and aims, which would then link the methods to the results more clearly.*

**Response**: Indeed, the manuscript was restructured according to the new aim outlined below in order to streamline the understanding of the presented ideas and facilitate the identification of the red thread. We acknowledge the extensive content of the methodology, and contracted it to 72.22% of its initial length (from 2517 to 1818 words), removing the subsection called Exploring the Impact Chain as well as other paragraphs in the other 2 subsections, and consequently amending the methodological workflow figure (Figure 3).

By removing one of the two initial objectives of the paper (the one relating to the exploration of multi-hazard impacts), the operation with two Impact Chains (one for the first objective, and one for the second) became unnecessary. Therefore, the paper relies on a single, enhanced Impact Chain, which builds on an initial version developed within the Paratus Project. All particular changes are highlighted by track changes in the manuscript.

Overall, we reduced the size of the paper to 78% of its initial size, from 13 252 words to 10 336 words (without references).

*Major comments*

*Introduction*

**R1:** *The introduction needs significant restructuring and rewriting to draw out the main research gap and aims that the paper is trying to close. The use of non-technical language makes it hard to follow. Review the use of sentences such as the following.*

*- The third decade of the 21st century debuted with a pivotal epidemiological hazardous event that taught human communities worldwide formative and often cruel lessons.*

*- In the new multi-hazard-prone era*

**Response:** In the reviewed version, the Introduction was restructured following the guidance of the reviewer in order to streamline the comprehension of ideas.
We modified the indicated phrases, as well as others that display the same shortcoming. Also, the entire manuscript was checked to remove such formulations.

The main research gap is extensively presented in two paragraphs, lines 82-94: *Up to date, scientific works on the interactions between natural hazards and the COVID-19 pandemic have primarily revolved around factual observations, overlooking the effects on the dynamics of vulnerability. Many examples (e.g., Andrews 2020, Majumdar and Dasgupta 2020, UNDRR 2020, Kassegn and Endris 2021, Mangubhai et al. 2021, Mishra et al. 2021, Patway and Rodriguez-Morales 2021, Pramanik et al. 2021, Izumi and Shaw 2022) pertain to hydro-climatic hazardous events amid the pandemic, offering only factual documentation on their interactions. Narrowing down to the flood hazard, the compounded impacts of flood events and the pandemic are largely unknown and have been described only tangentially or in short (Simonovic et al. 2020, Patway and Rodriguez-Morales 2021, Pramanik et al. 2021, Turay 2022), although the pandemic can augment typical health-related flood impacts (e.g., injuries, gastric problems stemming from water contamination, increased stress and/or anxiety) (Simonovic et al. 2020). Instead, more literature is available on the potential effects of flood events on the dynamics of COVID-19 cases (Frausto-Martínez et al. 2020, Mavroulis et al. 2021a, b, Albulescu 2023). What is more, the augmentation or attenuation of vulnerability conditions by previous hazard impacts (be they floods, pandemics, or other hazards) was not considered in any case study and has only been documented related to long-term processes (de Ruiter and van Loon 2022).*

This research gap-related paragraph is followed by the paragraph with the aim of the study (see below).

**R1:** *The main sentence and aim of the paper is hidden "This study delves deeper into the changes in vulnerability under hazard-generated impacts, taking as a case study two co-occurrent, independent hazards (i.e., floods and the COVID-19 pandemic) that severely affected a European country." and the current text does not speak so much to this. The introduction should be restructured to present the challenge, gap and how your work supports closing it.*

**Response:** The aim phrase starts the paragraph at line 95-99. It was contracted to focus on the augmentation of vulnerability by impacts and adaptation options in the proposed multi-hazard context: *This study aims to address the research gap regarding the dynamics of vulnerability in a multi-hazard context by analysing the increases in vulnerability that stem from hazard impacts and adaptation options, taking as a case study the co-occurrent extreme river flood events and the COVID-19 pandemic in Romania of 2020 and 2021.*

The Introduction was restructured as indicated by the reviewer: the challenge (lines 42-81), the gap (lines 82-94), the aim (95-108), and the contribution of the paper to the effort of reducing the gap (109-119). This structure is briefly presented at lines 42-50: *Given the increased frequency of co-occurrent or cascading hazards, vulnerability consolidated its key position in multi-risk analysis because the impact of multiple hazards and adaptive strategies reshaped its spatial and temporal dynamics. This raises significant challenges for risk management while reinforcing vulnerability's role in portraying disasters as human constructs (de Ruiter and van Loon 2022). This study delves deeper into the changes in vulnerability under hazard-generated impacts, taking as a case study two co-occurrent, independent hazards (i.e., floods and the COVID-19 pandemic) that severely affected a European country. At the outset, it is necessary to clarify the role of impacts resulting from multiple hazards in shaping vulnerability, with illustrative recent examples from the literature. These instances bring to light a notable research gap that requires investigation, as detailed in the following.*

**Methodology**

**R1:** *The number of different steps in the methodology make it very hard to follow. While interesting, it is a somewhat eclectic approach in certain areas. I do not agree with the comment "Elevating the Impact Chain from its above mentioned original purposes to a diagnosis and prediction tool represents a pioneering research endeavor, standing out as an element of methodological novelty". Rather, i feel that the entire paper should step away from the statement that you are predicting vulnerability dynamics, as it does not account for the myriad other factors that influence vulnerability (e.g. governance, development, systemic risks) etc etc.*

**Response:** We are thankful to the reviewer and agree that the methodology included too many steps and that the previous aim of the paper was extensive. By contracting the aim and simplifying the steps associated with the new objective, we removed 64% of the steps in the initial Methodology, and shortened this section by 27.78%. This is best illustrated in the new Figure 3. Methodological workflow.

We appreciate this suggestion of the reviewer, and we have also clarified the objective of the paper at lines 237-244 at the beginning of Methodology (and also in the Abstract): *The proposed methodological framework aims to identify and analyse the augmentation in vulnerability within a multi-hazard context. This framework dwells on Impact Chains as instruments for documentation, visualisation, organisation, and scientific inquiry, ultimately broadening their application to fit the latter objective of studying the dynamics of vulnerability – particularly the augmentation of vulnerability, and turning them into diagnosis and prediction tools. With this addition, the documentary focus of the chain progresses to a more analytical stance, specifically*

*geared towards identifying and tracking the transformation of specific vulnerabilities into drivers of vulnerability, thus becoming a tool for multi-hazard management in predicting potential crises due to deficiencies in management approaches.*

For an early clarification, we also ended the Introduction with a supplementary clarification (lines 109-119): *This research work makes a significant contribution to the field of DRR by broadening the original purpose of the Impact Chain, transforming it into a first-hand, semi-qualitative tool for analysing vulnerability. Through this expansion, the Impact Chain is elevated from a documentation tool to a diagnosis and prediction instrument. The focus is on advancing its application to delve into the intricate multi-hazard impacts, along with their ramifications on vulnerability conditions. The conceptual framework dwells on the argument of Otto and Raju (2023), who highlight that climate change should not be entirely blamed for climate-related disasters and that vulnerability conditions must be factored in when analysing impactful events. Placing greater emphasis on the vulnerability component brings up the necessity of understanding its dynamics across time and space (de Ruiter and van Loon 2022), and even more in multi-hazard situations. This can be achieved by expanding the scope of Impact Chains to give visibility to such shifts in vulnerability, to diagnose past or present multi-hazard risk management, and to predict potential crises, shortcomings of management approaches, and the transformation of certain vulnerabilities into drivers of vulnerability.*

The vulnerabilities included in the Impact Chain and in the manuscript include governance and development aspects. Some examples: improper governance structure for effective flood management, flood management not adapted to the COVID-19 context, ineffective institutional communication, development of inhabited areas in flood prone areas, development of infrastructure in flood prone areas, poverty, depleted capacity due to seasonal patterns of hazards, low quality construction materials, ineffective sewage system.
In order to eliminate any confusion, these vulnerabilities were highlighted at lines 537-542 in subsection 4.1 of the Results.

**R1:** *The ranking of vulnerabilities based on their augmentation is a step that you could consider removing from the publication, given its length and that it is trying to cover a lot for one paper.*

**Response:** The ranking of vulnerabilities based on their augmentation is part of the focus of the paper: to analyse the augmentation of vulnerability in a multi-hazard context. As we removed the aim and former Result subsection 4.1 (that dealt with the analysis of multi-hazard impacts within the Impact Chain), the length of the paper was drastically reduced.

*Results*

**R1:** *4.1 reads like a literature review. I do not see how it links to the methodology presented in the previous section. Are the impacts and events you are describing findings from the synthesis of literature and enhancing the impact chain? If so, state this. The main focus of the paper is how multi-hazard interaction and responses have*

*augmented vulnerability. I suggest to focus of this and reduce the other findings to keep the narrative more easy to follow.*

**Response:** Indeed, we revised and removed this section and focused the aim of the paper on the augmentation of vulnerability, as recommended by the reviewer. By doing so, the Results section was shortened and adapted to the changes implemented to Methodology.

**Discussion**

**R1:** *The discussion would benefit much more from reflecting on the methodology and its limitations. currently the limitations are mostly focused on data limitations, and not the limitations of the approach that you took, of which there are some significant ones. A discussion that looks at the novelty of the methods, and how they can be improved would significantly strengthen the paper.*

**Response:** We thank the reviewer for pointing out these shortcomings in the original version of the manuscript. We addressed the concerns by improving subsection 5.3. Limitations and constraints, adding the intrinsic limitations of the Impact Chain-based methodology from the perspective of the statistical approach (lines 133-136): *A notable methodological limitation refers to the lack of testing against other case studies and external validation; which we plan to address in the future by applying the methodological framework to other Impact Chains focusing on different multi-hazard case studies. Finally, the paper provides a limited view on the dynamic trajectory of vulnerability, relying only on two temporal pictures captured by the initial Impact Chain and the enhanced version of it.*

We will diligently take into account the recommendation of the reviewer, giving it careful consideration, and we will engage in thorough reflection on how we can improve the methodology in future papers. At this time, our interest is to centre the Discussion on the conceptual paths of increasing vulnerability, drawing exclusively from the events outlined in the enhanced Impact Chain. To further statistically test these conceptual paths is a future objective we intend to pursue in forthcoming research works.

The Discussion was modified to include a special subsection (5.2. Contribution and novelty) that describes the novelty of the methods and the contribution of the paper, as well as further steps we plan to implement to improve the proposed methodological framework. Changes were made between lines 97 and 117 to enhance readability.

**Conclusions**

**R1:** *Could be much sharper. The seven key take aways should be reduced to 2/3, that speak to the method you developed and the context specific findings form your case study.*

**Response:** To sharpen the outcome, we reduced the list of key takeaways from 7 to 5 specific findings worth keeping in mind by the reader, which were expressed in bullet form.

**R1:** *The statement that "Vulnerability is expected to increase due to inaction" is simply not true and is simplification of realty. interaction does not equal intensitication.*

**Response:** We generally agree with the reviewer, but we draw attention to a particularity of the Romanian society, expressed by facts summarised in the presented case study, in the Discussion section (lines 49-95). The absence of action has the potential to augment vulnerability because 1) the number of adaptation options is very low compared to the large number of vulnerabilities (13 to 26), meaning that many vulnerabilities are left unaddressed, 2) the adaptation options do not target multi-hazard, augmented vulnerabilities, but rather the impacts of the hazards (in the endeavour of addressing the symptoms of the crisis, and not its root causes), 3) the sequence of flood events and pandemic waves provides only a narrow timeframe to replenish capacity, which ends up depleted when facing the next flood/pandemic wave or even between flood events.

**Minor comments**

**Abstract**

**R1:** *Review the use of non-scientific language, which will make it more direct and easy to pull out the main messaging*

**Response:** The Abstract was rewritten removing the language indicated by the reviewer (page 1, lines 9-23.

**Setting the scene**

**R1:** *While the content is all relevant, i feel it can be shortened to get the message across more quickly.*

**Response:** We reduced the length of this section by 29.66%, from 1615 words to 1136 words.

Respectfully yours,
The Authors

---

## Author Comment (AC2)

**Response to Reviewer 2**

We are grateful to the reviewer for her observant attention and dedicated time invested in the thorough review of the manuscript. We dutifully engaged in responding to each of the meticulous comments in order to meet the rigorous academic requirements set forth by the reviewer.

Please find below the point-by-point responses. The line numbers correspond to the updated manuscript with track changes.

*The reviewers' comments are written in italics*, and our responses are in regular font.
*We chose blue and italic formatting for citations from the manuscript.*
All of the line numbers refer to the reviewed version of the manuscript.

**R2:** *The work presented in this manuscript delves into the challenging and forward-looking realm of assessing vulnerability dynamics arising from multi-hazard risks. It does so by introducing a novel application of Impact Chains within a multi-hazard framework. For these reasons, **this work holds significance and potential for publication following substantial restructuring.***

**Response:** We greatly appreciate the interest and positive feedback from the reviewer. Indeed, the manuscript was substantially restructured and contracted, as explained below. Overall, we reduced the size of the paper to 78% of its initial size and massively restructured the Methodology and Results sections according to the contraction of the aim. Furthermore, we revised the approach to center around a single Impact Chain (the enhanced version) to facilitate clarity and comprehension.

**R2:** *I report hereafter the major issues that I strongly encourage the authors to fix before publication:*

*The final aim of this work is not well explained in the introduction. Is the analysis of multi-hazard pandemic-floods vulnerability dynamics in Romania the main goal, or is it the development of an "enhanced" multi-hazard Impact Chain approach? The authors should pay more attention in framing their research question.*

**Response:** We clarified the aim of the paper at lines 102-109, contracting it to focus on the dynamics of vulnerability in the selected multi-hazard context. This goal was achieved by developing an enhanced Impact Chain. The confusion attentively noted by the reviewer was addressed at lines 102-109: *This study aims to address the research gap regarding the dynamics of vulnerability in a multi-hazard context by analysing the increases in vulnerability that stem from hazard impacts and adaptation options, taking as a case study the co-occurrent extreme river flood events and the COVID-19 pandemic in Romania in 2020 and 2021. The proposed methodological framework relies on an enhanced version of the initial Impact Chain developed within the Paratus Project (PARATUS Deliverable 1.1 202) to document the two-year unfolding of the two independent but co-occurrent hazards. This was upgraded to capture the shifts in vulnerability by enriching it with additional elements and connection types.*

**R2:** *Lines 93-94: It is not explained which specific transformation has been performed to the original Impact Chains approach. Clarifying this from the outset of the paper would be advantageous, as it would better underscore the novelty of the approach and its advancement beyond the current state-of-the-art. Additionally, it would be beneficial to provide a brief explanation of what are Impact Chains in the introduction, outlining their typical development purpose and traditional field of application.*

**Response:** We thank the reviewer for this valuable suggestion. We added a brief definition of Impact Chains in the Introduction (lines 110-113): *Impact Chains are conceptual models designed to visualise, document, and analyse the interconnections between hazards, vulnerability, and exposure that ultimately give rise to a specific risk (IPCC 2014, Zebisch et al. 2017). In this study, we refined the model to focus on the vulnerability dynamics in a multi-hazard context.*

This is followed by a paragraph with a presentation of the merits and novelty of this approach (lines 113-130). Furthermore, the state-of-the-art details are given in Methodology, at lines 274-287, and the enhancement of the chain is thoroughly presented in the new Section 3.2. Enhancing the Impact Chain.

**R2:** *The paper is very long and sometimes difficult to follow. I invite the authors to consider shortening some sessions. More specifically, I suggest shortening Sections 2 and 4.*

**Response:** Truly, the initial length of the paper hindered the following of the read thread and the easy understanding of the ideas. We thank the reviewer for having the patience to go through the paper and highlight what we have to change to make this a more pleasant experience for future readers. The size of the manuscript was reduced by 22%. Section 2 (Setting the scene) was shortened to 70% of its prior length, and Section 4 (Results) to 62% of its prior length. As the objective of the paper was contracted, Section 4.1. (presenting the multi-hazard impacts) was removed altogether.

**R2:** *Lines 152-154: warnings do not always result in actual floods. It's unclear how you linked warnings to real flood events. Did you incorporate data from other sources, as indicated in line 158? This aspect lacks clear elucidation. If mentioned, it should be elaborated upon, including details of the validation procedure.*

**Response:** This shortcoming was clarified at lines 183-186: *The flood events taken under analysis in this paper were identified using the hydrological warnings issued by the National Institute of Hydrology, and Water Management during 2020-2021, which were corroborated with information from a national news platform. Multiple news reports were used for the validation of each extracted piece of information.*

**R2:** *The methodology, as explained in Section 3 and illustrated in Figure 4, is very complex and difficult to understand and follow. More specifically, I do not understand the rationale behind presenting two distinct impact chains —the one from the PARATUS project and the one incorporating vulnerability dynamics, and to make a comparison among them (e.g., in Fig.7). In my view, this unnecessarily complicates the methodology and results presentation, potentially confusing the reader. Instead, it would be advantageous to emphasize the modifications made from the original Impact Chains approach to the one delineated in this paper. This should encompass alterations introduced to accommodate multiple hazards as well as those aimed at identifying patterns of dynamic vulnerability. I suggest restructuring the overall methodology to simplify it and also change the presentation of the results in Chapter 4 accordingly.*

**Response:** We align with the reviewer's opinion on the convoluted structure of the initial Methodology section. We restructured it according to the new aim (focusing only on analysing the dynamics of vulnerability), removing the former Section 3.2. Exploring the Impact Chain. The new approach relies only on the enhanced version of the Impact Chain, with the modifications made from the original presented at lines 283-288, and 368-406. Please note that we also simplified the methodological workflow figure (new Figure 3). By contracting the aim and simplifying the steps associated with the new objective, we removed 64% of the steps in the initial Methodology and shortened this section by 27.78%. The Results section was also modified to match the changes in Methodology.

**R2:** *Line 224: The Impact Chains approach is a standardized and codified procedure, well-developed and documented in a substantial body of literature. Given that this work builds upon the "original" Impact Chains approach, it is essential to provide additional background references on this methodology and introduce its founding principles in more detail. Regarding the references, please consider citing the following:*

> *Menk, L., Terzi, S., Zebisch, M., Rome, E., Lückerath, D., Milde, K., & Kienberger, S. (2022). Climate change impact chains: a review of applications, challenges, and opportunities for climate risk and vulnerability assessments. Weather, Climate, and Society, 14(2), 619-636.*

> *Zebisch, M., Schneiderbauer, S., Fritzsche, K., Bubeck, P., Kienberger, S., Kahlenborn, W., ... & Below, T. (2021). The vulnerability sourcebook and climate impact chains–a standardised framework for a climate vulnerability and risk assessment. International Journal of Climate Change Strategies and Management, 13(1), 35-59.*

> *Schneiderbauer, S., Baunach, D., Pedoth, L., Renner, K., Fritzsche, K., Bollin, C., ... & Ruzima, S. (2020). Spatial-explicit climate change vulnerability assessments based on impact chains. Findings from a case study in Burundi. Sustainability, 12(16), 6354.*

**Response:** We are very grateful to the reviewer for pointing out this issue and for providing us with the above-mentioned references. Details on previous applications of Impact Chains were included at lines 274-288, highlighting the differences between them and the current approach: *Impact Chains represent conceptual models designed to facilitate the investigation of climate and disaster risk under a structured analysis framework for the risks associated with climate-related impacts (UNDRR 2022). They have been used for elicitation, conceptualisation, analysis, and information sharing purposes, as tools that explore and analyse the impacts of single hazards or multi-hazards specific to past or potential hazardous events, following different operational frameworks (e.g., expert workshop, desktop analysis, machine-generated) and taking into consideration different spatial and temporal scopes (Pittore et al. 2023). There are numerous examples where Impact Chains were integrated into vulnerability or risk assessments specific to climatic aspects (Becker et al. 2014, Schneiderbauer et al. 2020, Zebisch et al. 2017, 2021, Menk et al. 2022).*

*In this paper, Impact Chains were used as models of cause and effect (Menk et al. 2022) that were upgraded to capture the augmentation of vulnerability by hazard impacts or adaptation options, with a limited participation of stakeholders (i.e., only integrating the feedback of first responders involved in flood emergency interventions).*

*Unlike the scientific papers reviewed by Menk et al. (2022), this study does not integrate Impact Chains as tools for the assessment of vulnerability or risk pertaining to a climatic hazard, but broadens their scope to focus on vulnerability dynamics within a multi-hazard context that involves a hydrological hazard (i.e., flood) and an epidemiological one (i.e., the COVID-19 pandemic). This approach aligns with Zebisch et al. (2021) recommendation that the "relatively linear and sectorial approach of impact chains could be widened to impact webs, which would include feedback relations and cross-connections."*

**R2:** *The "enhanced" Impact Chains approach introduced in this manuscript is presented in several sentences along throughout the text as a predictive tool. However, it appears that Impact Chains are primarily utilized as an analytical tool to deepen the understanding of multi-hazard vulnerability dynamics. I encourage the authors to reconsider this aspect and revise this concept throughout the manuscript.*

**Response:** We thank the reviewer for this tuned suggestion that we accordingly applied throughout the manuscript. The Impact Chain was upgraded to focus on the analysis of vulnerability dynamics and in the future we plan to further improve it and turn into a diagnosis and prediction tool (in another paper). In the reviewed manuscript, we addressed these concerns at the beginning of Methodology, at lines 252-257: *The proposed methodological framework aims to identify and analyse the augmentation in vulnerability conditions within a multi-hazard context. This framework dwells on Impact Chains as instruments for documentation, visualisation, organisation, and scientific inquiry, ultimately broadening their application to fit the objective of studying the dynamics of vulnerability – particularly the augmentation of vulnerability and henceforth to turn them into diagnosis and prediction tools. With this addition, the documentary focus of the chain progresses to a more analytical stance, specifically geared towards identifying and tracking the transformation of specific vulnerabilities into drivers of vulnerability.*

and also at the end of the Introduction, at lines 127-130: *This can be achieved by expanding the scope of Impact Chains to give visibility to such shifts in vulnerability, and further on to diagnose past or present multi-hazard risk management, and to predict potential crises, shortcomings of management approaches, and the transformation of certain vulnerabilities into drivers of vulnerability.*

**R2:** *Section 3.2 – "Exploring multi-hazard impacts" presents several unclear points.*

*How the "relevance" and "confidence" parameters are related to the identification of multi-hazard impacts?*

**Response:** As we contracted the aim of the paper to focus on the analysis of vulnerability dynamics, the Exploring multi-hazard impacts parts in Methodology and Results were deleted. Please note that these changes were also required by the first reviewer.

We decided to pursue the deleted objective (of analysing the multi-hazard impacts) in a forthcoming paper. The following insights from the reviewer are much appreciated, as they underscore the improvements we have to implement in the new planned paper, helping us to avoid future misunderstandings.

However, we want to briefly clarify the issues raised by the reviewer. The relevance and confidence parameters were assigned values from 1 to 10, with the highest scores pertaining to multi-hazard aspects. Thus, a high relevance to the multi-hazard coupled with a high confidence score indicated the multi-hazard impacts we had to focus our analysis on.

**R2:** *The "relevance" and "confidence" parameters are determined by applying the logical data model provided in Figure 5. However, interpreting this scheme may prove challenging for the reader without a more detailed explanation in the main body of the text. Please, explain more in detail.*

**Response:** The entire former Section 3.2. was deleted, and implementing the suggested changes in the current manuscript was thus rendered unnecessary. We noted this recommendation and will implement it in our future paper.

**R2:** *Please consider explaining the significance of the two metrics presented in Table 2 for identifying multi-hazard impacts. This could also be achieved by incorporating a dedicated column directly into the table.*

**Response:** We will consider this observation for the future paper. In this manuscript, the change is unnecessary because it no longer aligns with the ideas presented in the former Section 3.2. regarding the analysis of multi-hazard impacts.

**R2:** *The presentation of the Results in Section 4 does not follow the same structure of the methodology, as it is presented in Section 3, and this is not facilitating the understanding of the overall work. I suggest the authors simplify the methodology illustrated in Section 3 and then present the results accordingly.*

**Response:** Both the Methodology and Results sections were restructured and partially rewritten to match the contracted aim of the paper. The Methodology section was simplified and reduced to 72.22% of its initial length. Now, it is structured around the enhanced version of the Impact Chain, with 2 sections: the first presenting the building of the Impact Chain initially developed within the Paratus Project, and the second presenting the enhancements implemented to the Impact Chain to capture the various types of vulnerability augmentation. The Results section was shortened to 62% of its initial length, deleting the former Section 4.1. that referred to the multi-hazard impacts. Now, the Results focus only on the augmentation of vulnerability and align well with the new Methodology section.

**R2:** *Lines 400-401: It is still not clear to me how the Kumu metrics and the relevance parameters are combined to practically identify multi-hazard impacts. This aspect lacks clarity in Section 3 and remains somewhat ambiguous even when transitioning from the methodology to the results. The authors should provide a clearer and replicable explanation of their methodology, particularly concerning the novel aspects they have introduced.*

**Response:** We recognise the value of this observation and have noted it to amend in the future paper. In this manuscript, we removed the goal of analysing the multi-hazard impacts based on the relevance and Kumu parameters, the corresponding Section 3.2. in Methodology, and Section 4.1. of the Results. The new methodological framework highlights the novel aspects in the new Section 3.2. Enhancing the Impact Chain (lines 367-436). In addition, the new Figure 3. Methodological workflow should illustrate in a clearer, simplified way the main steps of the methodology.

**R2:** *Following from the previous point, I strongly recommend that the authors incorporate into the Appendix an integration of the original Impact Chains guidelines by Pittore et al. (2023). This would require presenting the new elements they have introduced to the methodology in a standardized and replicable manner.*

**Response:** We thank the reviewer for this suggestion and have carefully discussed it. However, including the guidelines by Pittore et al. (2023) seems to bring little value to

the manuscript, as they are readily accessible through the provided reference. Including them would increase the length of the paper and introduce a didactic tone that we would like to avoid.

**R2:** *Section 5.2: I invite the authors to discuss also some methodological limitations or assumptions they made, which are currently not mentioned, e.g. the limited participation of stakeholders in the construction of the Impact Chains. Indeed, Impact Chains are well suited for use in a transdisciplinary perspective for the co-production of knowledge.*

**Response:** We profoundly reflected on this comment and agree with the reviewer. The participation of stakeholders is limited to the feedback offered by first responders who performed on-site interventions during the flood events of 2021. We noted this limitation in the new Section 5.3. Limitations and constraints, together with other 2 methodological limitations, at lines 134-142: *The implication of stakeholders in the construction of the multi-hazard Impact Chain is limited to the feedback provided by first responders who performed on-site emergency interventions during the floods of 2021 (Fekete et al. 2023). Future research directions focus on a broader involvement of different stakeholders in order to maximise the benefits of co-produced knowledge and refine the details specific to the multi-hazard context from a transdisciplinary perspective. A notable methodological limitation refers to the lack of testing against other case studies and external validation; which we plan to address in the future by applying the methodological framework to other Impact Chains focusing on different multi-hazard case studies. Finally, the paper provides a limited view on the dynamics of vulnerability, relying only on two temporal pictures captured by the initial Impact Chain and the enhanced version of it. Some of these methodological limitations are inherent to Impact Chain-based analyses, as highlighted in the literature review performed by Menk et al. (2022).*

**R2:** *The entire manuscript would benefit from a systematic review of the language, including enhancements to the vocabulary and terminology.*

**Response:** During the review process, we reread and rewrote large parts of the manuscript, amending all of the identified language errors and vocabulary/terminology-related shortcomings. With the help of reviewers, the new version of the manuscript is improved in this regard too.

**Other medium to minor issues:**

**R2:** *Lines 30-31: The authors affirm that the co-occurrence of COVID-19 and other natural hazards has "caused a paradigm shift" from multi-ayer single hazard approaches to interacting hazards. Indeed, the shift has been started before the occurrence of COVID-19. The co-occurrence of COVID-19 and other natural hazards has instead increased the attention to potential synergies and asynergies between pandemics and other hazards from the Disaster Risk Management and Emergency Management perspectives (see Terzi et al, 2022). Please, consider rephrasing this sentence.*

**Response:** Yes, we acknowledge the nuance highlighted by the reviewer. We rephrased the idea (lines 33-38) as: *In the field of Disaster Risk Reduction (DRR), the co-occurrence of natural hazards of various types and magnitudes amid the COVID-19 pandemic has increased attention to potential synergies and asynergies between pandemics and other hazards (Terzi et al. 2022). Even before the pandemic, multi-*

*hazard analysis switched its focus from analysing all the hazards that can affect an area in a given period of time, which is often called multilayer single hazard analysis (Gill and Malamud 2014) or "all-hazards-at-place approach" (Hewitt and Burton 1971), to analysing the interactions between the hazards that overlap in time and space (De Angeli et al. 2022).*

**R2:** *Lines 34-35: please add references for the Sendai Framework and the Paris Agreement.*

**Response:** We provided the references for the Sendai Framework and the Paris Agreement at line 39.

**R2:** *Lines 42-43: "a European country". I think it would be more beneficial to indicate directly which country.*

**Response:** Yes, we clarified that the European country in question was Romania (line 50).

**R2:** *Lines 61-62: Here there is a sudden shift of topic. The authors previously discussed vulnerability dynamics up to line 61, but then abruptly shifted focus to operational hazard management procedures during the COVID-19 pandemic. It seems there's a gap here, lacking a sentence that introduces why and how COVID-19 has posed challenges from a multi-hazard perspective.*

**Response:** We thank the reviewer for helping us improve the flow of the Introduction. Indeed, we identified the missing link between the two ideas. The shortcoming was amended by adding the following connecting phrases at lines 69-72: *The interactions between the COVID-19 pandemic and co-occurrent natural hazards add layers of complexity to analysing vulnerability dynamics and constructing DRM models that factor in this dynamics. The complications arise from the necessity to adjust traditional natural hazard management approaches to the new pandemic conditions, with implications for both the impacts and the adaptation options that can increase vulnerability.*

**R2:** *Table 1. This table, as it is presented, is not very informative and useful. It would be more beneficial to add a column to specify which kind of management issue has been encountered in each of the presented real-world cases.*

**Response:** We deleted Table 1 in order to shorten the Introduction (as requested by the first reviewer). The references from the table were included in the paragraph at lines 73-79, which also highlights the overall management issues depicted in the case studies referred to: *Scientific literature provides several examples (Andrews 2020, Majumdar and Dasgupta 2020, UNDRR 2020, Kassegn and Endris 2021, Mangubhai et al. 2021, Mishra et al. 2021, Patwary and Rodriguez-Morales 2021, Pramanik et al. 2021, Izumi and Shaw 2022) that point out failures of hazard management, which stem from the fact that standard operational procedures were not adapted to pandemic conditions, or from the fact that the efforts of tilting the SARS-CoV-2 infection curve were not adapted to fit hazard management practices. In recent years, this conundrum has become a hot topic in the field of DRM, being debated by numerous scientists (Frausto-Martínez et al. 2020, Quigley et al. 2020, Potutan and Arakida 2021, Albulescu et al. 2022, Hariri-Ardebili et al. 2022).*

**R2:** *Section 4.1, lines 378-381: several hazards are mentioned, but only pandemics and floods are selected. The authors should support this choice by providing some evidence. Moreover, I suggest moving this discussion to the beginning of Section 2, when the case study of floods and COVID-19 is discussed for the first time.*

**Response:** According to the new aim, we deleted Section 4.1. which detailed the multi-hazard impacts. We value the suggestions on how to improve Section 2 by motivating the choice of the flood and COVID-19 hazards. We addressed this by introducing the following phrase at lines 133-135 in the beginning of Section 2: *Floods, the COVID-19 pandemic, and heavy rainfall were considered primary hazards within the Impact Chain, but only the first two are analysed in this study due to their significant impacts. Other secondary hazards (e.g., strong wind, landslides) co-occurred with the other two, but their role was of lesser significance in the analysed multi-hazard context.*

The motivation for the selection is presented at lines 152-158: *Floods are among the most common and impactful natural hazards that affect Romania, causing significant damage throughout the country. The EM-DAT (2023) database includes 102 natural hazardous events that occurred in Romania in 1900-2023, of which flood events represent almost 52%. These floods resulted in more than 1700 deaths, more than 146600 homeless people, over 1.64 million affected people, and total estimated damages of about 8.69 billion dollars. This incomplete dataset, complemented by other European flood-related databases (e.g., HANZE v2.1 developed by Paprotny and Mengel 2023, Paprotny et al. 2023) points out the prominence of floods among the natural hazards that occur in the country of reference.*

and at lines 213-217: *Until the beginning of June 2023, more than 3.4 million cases of COVID-19 and over 68,000 deaths were registered in the country of interest, of which 53.07%, respectively 86.09% can be traced back to the first two pandemic years (WHO Dashboard 2023). The largest number of both COVID-19 cases (1,179,282) and COVID-19-induced deaths (43,118) occurred in 2021. This human toll unfolded in five pandemic waves (Figure 32), of which the fourth one, starting in 2022, was the most aggressive.*

**R2:** *Section 4.1: it would be beneficial to include a table summarizing the main multi-hazard impacts discussed in the text.*

**Response:** Given that Section 4.1. was completely deleted (as it no longer matched the contracted aim of the paper), the inclusion of a new table can be viewed as unnecessary. Nevertheless, we noted the useful recommendation and will include it in the future paper that will integrate the text of former Section 4.1.

**R2:** *Figure 8 does not seem to be so relevant to be included in the paper, since a specific analysis and discussion of spatial distribution of flood impacts is not performed in the study.*

**Response:** We acknowledged the validity of the comment and carefully considered the utility of the mentioned figure. The former Figure 8 was transformed into Figure 1, and placed at the end of Section 2.1. which presents the flood risk and hazardous events in Romania. At this point, we think that an illustrative presentation of the spatial distribution of the flood impacts is required, since Section 4.1. (which presented the multi-hazard impacts) was deleted. This substantial removal left the reader with little information on the severity of the flood hazards that affected Romania in 2020-2021, which can be partially addressed by the new Figure 1.

Respectfully yours,
The Authors

---

## Referee Report (RR1)

NHESS-2024-5 | Research article
**An Impact Chain-based exploration of multi-hazard vulnerability dynamics. The multi-hazard of floods and the COVID-19 pandemic in Romania**
*Andra-Cosmina Albulescu  and Iuliana Armas*

**Review – 2nd round**
* * *
**I sincerely thank the authors for kindly replying to all my comments and putting valuable effort into improving the readability of their manuscript. The paper is now clearer and generally easier to read and understand.**
Nevertheless, I still recommend fixing the following aspects before publication:

**1. Introduction.** Despite the implemented changes, the narrative of the Introduction remains fragmented. Lines 22-32 introduce the challenges of the COVID-19 pandemic. Then, the focus shifts to the evolution of multi-(hazard)-risk approaches and the related terminology, followed by vulnerability and its dynamic nature. After several lines, the discussion returns to COVID (line 62). Furthermore, the paper's aim is presented in a disjointed manner (lines 42-44, then lines 92-94, and finally 107 onwards). I suggest the authors reorder the concepts and present them in a more linear and logical sequence. This will not only enhance readability but also strengthen the presentation of the quality and added value of the work.

**2. Setting the scene.**
- I suggest modifying the title of Section 2.1 to "Flood risk occurrences in Romania".
- In lines 119-120, the authors state, *"Other secondary hazards (e.g., strong wind, landslides) co-occurred with the other two, but their role was of lesser significance in the analyzed multi-hazard context."* It would be beneficial for the authors to provide further clarification. Does "lesser significance in the analyzed multi-hazard context" imply that these secondary hazards were less impactful compared to the other hazards under consideration?
- Line 146, *"Flood risk management is not sufficiently documented in Romania, as demonstrated by the lack of databases regarding the occurrence and impacts of floods"*. I think it would be more appropriate to avoid mentioning risk management here and rephrase it as follows: *"Flood occurrences and the quantification of their associated impacts are not sufficiently documented in Romania, as evidenced by the absence of relevant official databases."*

**3. Methodology.** Thanks to the restructuring performed by the authors, this part is now much easier to read and understand.  To further improve it, I suggest clarifying the following:
- It is not so immediately clear to the reader if the authors built their impact chain from zero following the methodology developed in the Paratus Project, or if they used as a starting point an Impact Chain developed in the Paratus project but based on the methodology developed outside the project by Eurac. I mainly refer

to this unclear sentence: *"Building the Impact Chain initially developed within the Paratus Project"* (lines 201-202). I invite the authors to clarify this aspect from the beginning.

- The synthetic explanation of the Impact Chain construction in Section 3.1 lacks mention of all the steps depicted in Fig. 3. Furthermore, even when these steps are referenced, the text doesn't consistently use the labels that appear in the figure. To enhance the clarity of the methodology, I encourage the authors to ensure better alignment between the text in Section 3.1 and the content depicted in panel 1 of Figure 3.
- Section 3.2 would benefit from minor restructuring to enhance reader comprehension of the steps outlined. The authors wrote *"This broadening of the original application of the Impact Chain was done by 1) introducing new types of elements (i.e., augmented vulnerabilities, derived impacts), 2) establishing new types of connections between the impacts/adaptation options and vulnerabilities, and 3) ranking the vulnerabilities in the Impact Chain based on their augmentation. These steps were implemented to construct an enhanced Impact Chain, building on the previous version that documented the unfolding of the selected co-occurrent hazards in Romania in 2020-2021."*, but then, a few lines later, they start saying that *"The first step was to perform an in-depth analysis of the vulnerabilities in the Impact Chain."*. To ensure coherence, it is essential to present the steps in the correct order, and, also in this case, to refer more to the steps reported in Fig.3. Additionally, it would be helpful to introduce and define the new types of elements (i.e., augmented vulnerabilities, derived impacts) earlier in the text, perhaps before discussing the new types of connections. This anticipatory approach can enhance clarity and understanding for the reader.

**4. Results**

- Lines 398-401: "Some vulnerabilities underwent multiple transformations into derived impacts [...] The explanation lies in the fact that multiple impacts can augment the same vulnerability, creating also a derived impact that reinforce the impact that generated the augmentation." The concept expressed in these sentences is not so easily understandable. I invite the authors to consider rephrasing it.
- Despite "the intricate configuration of the Impact Chain does not allow for a proper visualisation within this paper", I think that it would be highly beneficial, even essential, to include some graphical representations of the derived impacts described verbally in Section 4.3. This section constitutes the core of the work and would be communicated more clearly and effectively by introducing some 'extracts' from the Impact Chains to aid in visualization.

Moreover, I strongly suggest the authors double-check **grammar and typos**.

---

## Author Response (AR2)

**Response to Reviewer 1 – Second round of revision**

We thank the reviewer for her/his/their appreciation, for following through with this review in the second round, and for providing us with more suggestions to improve the paper. We are grateful for all the provided support and remain dedicated to further refining the paper.
Please find below the point-by-point responses.

*The reviewers' comments are written in italics*, and our responses are in regular font.
*We chose blue and italic formatting for citations from the manuscript.*
All of the line numbers refer to the version of the manuscript reviewed in the second round.

**R1:** *I thank the authors for their consideration of my and my fellow reviewers' comments. I feel that the paper has become significantly sharper and improved a lot. I found the narrative and 'red thread' much easier to follow, and I believe that by reducing the papers objectives, it now does not aim to present too much work. The introduction, while I feel still needs some adjustments, is more focused and the research gap comes out more clearly. The setting the scene is now a much more appropriate length and to the point, providing enough and relevant information. The methodology is now clearer and the results are presented more strongly by reducing the number of tables and figures. This discussion has improved. I still feel the paper needs another round of revisions before it can be published in NESS. Find below my comments on specific sections.*

**Abstract**
*'Within multi-hazards, both the impacts of hazards and the mitigation strategies can augment vulnerabilities, adding layers to the complexity of multi-risk assessments.' - This statement is also true in a single hazard context, additionally, what is meant by within multi-hazards? Suggest to change to 'in a multi-hazard context'*
**Response:** Because of the word count constraints specific to the Abstract, we chose not to further define multi-hazard, but to change the phrase as per the reviewer's thoughtful recommendation (line 11).

**Introduction**
**R1:** *'Given the increased frequency of co-occurrent or cascading hazards, vulnerability consolidated its key position in multi-risk analysis because the impact of multiple hazards and adaptive strategies reshaped its spatial and temporal dynamics.' – This is a rather strange formulation, as vulnerability is already key in single risk analysis. I also do not fully understand what is meant by impacts and adaptation reshaping spatial and temporal dynamics of vulnerability. This needs unpacking. I would also adjust the language in the introduction as its currently framed as if vulnerability itself has consolidated its position or done something, it should be written in the third person.*

**Response:** We appreciate this suggestion. The indicated phrase was reformulated in the third person (lines 60-62), and the unpacking of the meaning is detailed at lines 68-82.

**R1:** *Review use non-scientific language including words and phrases throughout the paper such as 'tall order', 'conundrum', 'hot topic' 'dwells on impact chains'*

**Response:** We took into consideration this recommendation of the reviewer and we have made adjustments accordingly by reducing the usage of such language throughout the paper. Nevertheless, some idioms in the English language were kept, as they are largely accepted in scientific publications.

**R1:** *Lines 113 – 114: 'Such efforts are vital for elaborating post pandemic update risk management plans' – Remove 'Update'*

**Response:** The modification was implemented (line 92) as kindly suggested by the reviewer.

**R1:** *The paragraph between lines 120 – 130 is more of a conclusion to your work and should not be in the introduction.*

**Response:** After careful consideration about this suggestion, we decided to keep the paragraph in place, as it highlights the contribution of the paper to the field of research and rounds out the Introduction. We acknowledge the argument of Otto and Raju (2023) of utmost importance for motivating the need for such studies and the conceptual framework presented in the manuscript. Please note that the Introduction was restructured based on the recommendations of the second reviewer.

**Setting the scene**
**R1:** *The opening sentence here (lines 133 – 135) is not well places and should be shifted down to a more appropriate position*

**Response:** The indicated opening sentence was moved to lines 331-335, as part of the methodology section 3.1. Building the Impact Chain.

**R1:** *Figure 1 has merit and is interesting to show, however can the formatting/ size be adjusted so it is larger and more clear?*

**Response:** We thank the reviewer for the attention dedicated to this detail. The resolution of Figure 1 was increased to 700 dpi in order to make it clearer. Also, if possible, we will kindly ask the editorial staff to increase its size and place it on a whole page.

**Methodology**
**R1:** *Lines 286 - 288: "This approach aligns with Zebisch et al. (2021) recommendation that the "relatively linear and sectorial approach of impact chains could be widened to impact webs, which would include feedback relations and cross-connections." -*

*Suggest to include the following citation for this statement: Sparkes. E., Hagenlocher, M., Cotti, D. et al. (2023). Understanding and characterizing complex risks with Impact Webs: a guidance document. UNU-EHS, Bonn, Germany*

**Response:** We thank the reviewer for this reference, which was included as support for the argument at line 313.

**R1:** *Line 313 + Figure 3: What are on-point examples? It would be helpful to explain this*

**Response:** We formulated this phrase to be clearer, as we did not intend to highlight any particular examples (lines 337-339): *The first phase of the building process (Figure 3) relied on a literature review regarding the impacts of flood events and the pandemic, complemented by a supplementation of examples specific to the flood hazardous events in 2020-2021 collected from studying the grey literature.*
The elements of the Impact Chain that were extracted from grey literature sources (i.e., news reports) are visible in the Impact Chain on Kumu. We also revised Sections 3.1 and 3.2 to improve the flow of the explanations and to ensure that all methodological steps correspond to Figure 3. This revision amends the confusion indicated by the reviewer.

**R1:** *Line 335: 'Cumulatively, the Impact Chain drew from 46 scientific papers (including one on the feedback of first responders), one legislative document, one official press release, one Eurostat statistical dataset, 6 official reports, and 75 news reports.' – it also drew on the feedback from 595 first responders as well didn't it? I would include this knowledge source in this sentence as well, as it is very important*

**Response:** We thank the reviewer for pointing out this important aspect that was unintentionally left out. The information was added at line 364 and detailed in the next paragraph (368-373).

**R1:** *Lines 411-412: "Within the new conceptual framework of the enhanced Impact Chain (Figure 4), certain augmented vulnerabilities stand out also as impacts that deepen the impact that increased the vulnerability in the first place" – this sentence is hard to understand, how is an augmented vulnerability also an impact that deepened an impact that increased the initial vulnerability? Are you talking about feedback effects here? How is the 'sharpens' connection different from a feedback effect. I suggest to review the paragraph on lines 411 – 419 and rewrite so as it is more clear and understandable.*

**Response:** We greatly appreciate the reviewer for highlighting this misunderstanding. Please note that the indicated paragraph was revised to enhance clarity and understanding (lines 418-428): *Within the new conceptual framework of the enhanced Impact Chain (Figure 4), certain vulnerabilities, upon augmentation by an impact, can also act as impacts that further on deepen the very impact that increased the vulnerability in the first place. This process represents a positive feedback loop, where the initial impact augments a vulnerability that can be viewed afterwards as a (derived)*

*impact that will reinforce the first impact in the future. Such augmented vulnerabilities that also act as impacts were introduced in the enhanced Impact Chain as derived impacts and linked to the vulnerability element that they share their name with by "relates to" connections. These "relates to" links are not visible within the enhanced Impact Chain in Kumu in order to reduce the visual strain. Subsequently, the derived impacts were linked with the impact that deepened/shifted the corresponding vulnerability by a newly introduced type of connection referred to as "sharpens" (Figure 4). These "sharpens" connections convey the message that the augmented vulnerability will intensify the impact that initially augmented the vulnerability, rendering this impact more prominent than before.*

**R1:** *I feel it would be good to distinguish somewhere how you classified something as an impact and something as a vulnerability, as in the results I feel there is some overlap between the two. This would help the reader with clarification of statements on line 627 such as "When augmented, certain vulnerabilities act like impacts and reinforce the impact that increased the vulnerability in the first place."*

**Response:** The classification of certain elements in the Impact Chain as impacts or vulnerabilities was based on the guidelines provided by UNDRR (2017) and by Pittore et al. (2023). This was clarified at lines 315-326: *The structure of an Impact Chain includes elements that can be considered the fundamental units of a hazard-related context and the connections established between them. The elements can take the form of hazards, impacts, exposed elements, vulnerabilities, and adaptation options, defined according to the Sendai Framework Terminology on Disaster Risk Reduction (UNDRR 2017). Given the central role of impacts, vulnerabilities, and adaptation options in the proposed vulnerability augmentation framework, we consider that their meaning should be highlighted here. In this paper, impacts particularly refer to the negative effects of a hazardous event or a disaster, while vulnerability represents the "conditions determined by physical, social, economic and environmental factors or processes which increase the susceptibility of an individual, a community, assets or systems to the impacts of hazards" (UNDRR 2017). Adaptation options are measures meant to attenuate the negative impacts by addressing one or more vulnerabilities or impact mechanisms (IPCC 2014a). These elements are organised in a chain-resembling structure that relies on different connection types: causes, affects, relates to, impacts, and mitigates. Detailed guidelines on how such connections were established within the Paratus Project are provided by Pittore et al. (2023) and PARATUS Deliverable 1.1 (2023).*
In addition, please note that the statements at lines 627 of the previous version of the manuscript were rewritten for increased clarity (lines 418-428).

**Results**
**R1:** *Lines 442 – 445: "This section focuses on the augmentation of vulnerability stemming from certain flood or pandemic impacts and of the adaptation options implemented to mitigate vulnerabilities and/or impacts" – Here you separate flood and pandemic impacts, but in the problem statement you say you are looking at co-occurring and compounding impacts.*

**Response:** We thank the editor for highlighting this nuance. Indeed, we are looking at co-occurring and compounding events (which are all included in the Impact Chain). The phrase was modified to eliminate the confusion (lines 447-448).

**R1:** *Lines 557 – 559: "Most of the vulnerabilities contribute to prominent multi-hazard impacts such as the flooded/damaged houses or households, the flooded/damaged/blocked roads, the displaced/(self-) evacuated people, increased stress or anxiety, and the potential increase in COVID-19 new cases" – these are not all multi-hazard impacts, the first three are flood hazard impacts, increased stress/ anxiety could be considered multi-hazard, and increase in COVID-19 cases multi-hazard if you are explicit in saying flooding resulting in less social distancing, resulting in increased COVID cases, in which case it would be a cascading impact of flooding. I suggest you review here.*

**Response:** Indeed, the first three impacts are flood-related, and the last two are multi-hazard impacts. The potential increase in COVID-19 new cases was considered a multi-hazard impact, as it results from reduced social distancing and evacuation of the population because of the floods (please see the Impact Chain on Kumu for more details on this last impact or Albulescu 2023). We appreciate this thoughtful suggestion and revised it accordingly (lines 456-458).

**R1:** *Line 575: "When it comes to adaptation options, only 30.76% of the vulnerabilities were mitigated by such elements" – Do you mean adaptation options mitigated the effects/ were targeted to 30% of the vulnerabilities identified in the Impact Chain?*

**Response:** Yes, this is the intended meaning of the phrase. To resolve the confusion, the phase was rewritten (lines 473-477). We are very grateful for this thread of meticulous observations.

**R1:** *I do not think its necessary to present percentiles to two decimal places, e.g.: ", 27.77% to the COVID-19 pandemic, and 22.22% to both hazards." Consider just saying 28% and 22%, and integrate this throughout.*

**Response:** The percentages were rounded up as suggested by the reviewer throughout the paper.

**R1:** *Lines 629 – 634: "Some vulnerabilities were transformed into derived impacts more than once underwent multiple transformations into derived impacts, resulting in a larger number of cases where the augmentation of a vulnerability created a derived impacts (15 cases), compared to the number of actual derived impacts (9) in the chain. The explanation lies in the fact that multiple impacts can augment the same vulnerability, creating also a derived impact that reinforce the impact that generated the augmentation." – I find this formulation very hard to understand and convoluted. I strongly suggest to look throughout the paper again when describing this effect/ phenomenon you have observed in your results and explain it in another way that is more clear and easy to understand.*

**Response:** We are grateful to the reviewer for indicating these confusions that we amended throughout the paper (lines 418-428, 522-528). Section 4.3 was revised and modified to facilitate the understanding of the mechanism of derived impacts. It was also improved in terms of visual support by introducing a new figure (Figure 6).

The new paragraph reads: *When augmented, certain vulnerabilities can function similarly to impacts, reinforcing the very impact that initially increased the vulnerability, forming a positive loop feedback composed of "deepens/shifts" links and "sharpens" links. Such augmented vulnerabilities with double status were duplicated in the enhanced Impact Chain and labelled as "derived impacts", as detailed in Appendix B. Some vulnerabilities underwent multiple transformations into derived impacts because they acted as (derived impacts) in relation to more than one augmentation-generator impact (Figure 6). This resulted in a larger number of cases where the augmentation of a vulnerability created a derived impacts (15 cases) compared to the number of actual derived impacts (9) in the chain.*

**R1:** *Line 685: "while the low-performance medical system is specific to the pandemic" – Is it? The medical system would have also been put under strain from those injured from floods, suggest to be specific here if talking about performance for COVID-19 treatment*

**Response:** We agree with this keen observation of the reviewer on a general basis. However, this is not the case, as no flood injuries were reported in 2020 and 2021 in Romania. Therefore, the low-performance medical system is a vulnerability specific to the pandemic.

**R1:** *Line 681: "with the goal of pinpointing those vulnerabilities expected to experience the most substantial increase" – I am still struggling here to see how your statistical analysis can be used as a projection for expected future increase. When you say increase, do you mean those vulnerabilities that were most influenced by the past event of an extreme flood co-occurring with the COVID-19 pandemic, which can then point to where risk management and preventative interventions should be targeted for future? I feel this framing would better suit what you are showing with your work.*

**Response:** We thank the reviewer for pointing out the need to better highlight the role of the ranking procedure. This procedure for identifying the most augmented vulnerabilities is presented at lines 431-445. By building augmentation links between impacts and vulnerabilities (i.e., deepens, shifts) and between adaptation options and vulnerabilities (i.e., rebounds, creates negative externalities), we can identify which vulnerabilities are expected to increase in the future and because of what causes (for this, please see Section 5.1. Conceptual paths of rising vulnerability). The ranking of the augmented vulnerabilities is presented in the dedicated Section 4.4. Ranking of augmented vulnerabilities.

By increase in vulnerability, we mean that those levels of vulnerability are expected to be higher in the future, providing the next hazardous events will lead to similar impacts, and that similar adaptation options will be implemented to mitigate those vulnerabilities.

This explanation was added at lines 580-585.

**Discussion**

**R1:** *"Interest in vulnerability dynamics has surfaced since 2020 0, and discussions have remained at a theoretical level (de Ruiter and Van Loon 2022), with no case study up to date."* – This is incorrect, there has been interest in vulnerability dynamics for a far longer time that 4 years, and many case studies on it.

**Response:** We thank the reviewer for correcting this information. In the manuscript, it was amended as (lines 98-100): *Although vulnerability dynamics has gained traction over the last decades, interest in vulnerability dynamics within multi-hazard contexts has surfaced since 2020, and discussions have remained at a theoretical level (de Ruiter and Van Loon 2022), with no case study up to date.*

**R1:** *"This improved version of the chain"* – I would strongly avoid using this kind of framing, your adjustments to the Impact Chain method are relevant for your own work and research context here, they do not necessarily improve impact chains per se. This very much depends on what you want to achieve.

**Response:** We thank the reviewer for drawing attention to this improper formulation. The improvement refers mainly to the initial version of the Impact Chain (developed within the Paratus project). However, adapting the method to fit the purpose of analysing vulnerability augmentation can only be considered an enhancement of the method in that specific context.

**R1:** *"The dual functionality highlights the capability of the methodological framework to account for both changes in vulnerability and the intricacies of multi-hazard impacts"* – What do you mean by intricacies of multi-hazard impacts? Interconnectivity?

**Response:** Yes, this is the intended meaning. To increase clarity, we replaced "intricacies" with "interconnectivity" (line 113).

**R1:** *"Finally, the paper provides a limited view on the dynamics of vulnerability, relying only on two temporal pictures captured by the initial Impact Chain and the enhanced version of it."* – Unless I am misunderstanding, in your results do you not suggest that the enhanced impact chain can show vulnerabilities expected to experience the most substantial increase, thus as a predictive tool? If I have grasped this correctly your limitations contradict your results here.

**Response:** We thank the reviewer for drawing attention to this unclear formulation. Our intention was to suggest that developing Impact Chains with the same multi-hazard context for multiple years and tracking the progression of vulnerability augmentation along multiple moments in time (more than two temporal pictures) would represent a more refined and comprehensive approach. This is our plan for future research works that will build upon the findings presented in this paper. This issue was clarified at lines 140-142: *In the future, the development of Impact Chains within the same multi-hazard context but for multiple years, and the tracking of the augmentation of vulnerability across multiple temporal snapshots will yield more nuanced results that can also be validated with narratives from grey literature.*

**R1:** *I also feel a deeper reflection on the limitations of your methodological, in particularly your classifications of vulnerabilities and impacts, and your statistical analysis to rank vulnerabilities would be more appropriate in section 5.3. rather than reflecting mostly on data limitations.*

**Response:** We thank the reviewer for this suggestion. The methodology-related limitations presented in Section 5.3. are (lines 133-144): *The implication of stakeholders in the construction of the multi-hazard Impact Chain is limited to the feedback provided by first responders who performed on-site emergency interventions during the floods of 2021 (Fekete et al. 2023). Future research directions focus on a broader involvement of different stakeholders in order to maximise the benefits of co-produced knowledge and refine the details specific to the multi-hazard context from a transdisciplinary perspective. A notable methodological limitation refers to the lack of testing against other case studies and external validation; which we plan to address in the future by applying the methodological framework to other Impact Chains focusing on different multi-hazard situations. Finally, the paper provides a limited view on the dynamics of vulnerability, relying only on two temporal pictures captured by the initial Impact Chain and the enhanced version of it. In the future, the development of Impact Chains within the same multi-hazard context but for multiple years, and the tracking of the augmentation of vulnerability across multiple temporal snapshots will yield more nuanced results that can also be validated with narratives from grey literature. Some of these methodological limitations are inherent to Impact Chain-based analyses, as highlighted in the literature review performed by Menk et al. (2022).*

Respectfully yours,
The Authors

**Response to Reviewer 2 – Second round of revision**

We are greateful to the reviewer for her positive feedback, thorough engagement in the second round of review, and for the new valuable additional recommendations. We deeply appreciate all the support and futher commit to integrate the provided insights to enhance the quality of our research work.
Please find below the point-by-point responses.

*The reviewers' comments are written in italics*, and our responses are in regular font.
*We chose blue and italic formatting for citations from the manuscript.*
All of the line numbers refer to the version of the manuscript reviewed in the second round.

**R2:** *I sincerely thank the authors for kindly replying to all my comments and putting valuable elort into improving the readability of their manuscript. The paper is now clearer and generally easier to read and understand.*
*Nevertheless, I still recommend fixing the following aspects before publication:*

**1. Introduction.**
**R2:** *Despite the implemented changes, the narrative of the Introduction remains fragmented. Lines 22-32 introduce the challenges of the COVID-19 pandemic. Then, the focus shifts to the evolution of multi-(hazard)-risk approaches and the related terminology, followed by vulnerability and its dynamic nature. After several lines, the discussion returns to COVID (line 62). Furthermore, the paper's aim is presented in a disjointed manner (lines 42-44, then lines 92-94, and finally 107 onwards). I suggest the authors reorder the concepts and present them in a more linear and logical sequence. This will not only enhance readability but also strengthen the presentation of the quality and added value of the work.*

**Response:** The Introduction was rewritten according to the suggestions of the reviewer and enhanced for clarity.

**2. Setting the scene.**
**R2:** *I suggest modifying the title of Section 2.1 to "Flood risk occurrences in Romania".*

**Response:** We appreciate this recommendation, and decided to simplify by renaming Section 2.1 as "Flood Risk in Romania".

**R2:** *In lines 119-120, the authors state, "Other secondary hazards (e.g., strong wind, landslides) co-occurred with the other two, but their role was of lesser significance in the analyzed multi-hazard context." It would be beneficial for the authors to provide further clarification. Does "lesser significance in the analyzed multi-hazard context" imply that these secondary hazards were less impactful compared to the other hazards under consideration?*

**Response:** Yes, this is the intended meaning. This was clarified at lines 331-335.

**R2:** *Line 146, "Flood risk management is not suficiently documented in Romania, as demonstrated by the lack of databases regarding the occurrence and impacts of floods". I think it would be more appropriate to avoid mentioning risk management here and rephrase it as follows: "Flood occurrences and the quantification of their associated impacts are not suficiently documented in Romania, as evidenced by the absence of relevant oficial databases."*

**Response:** The phrase was modified according to the instructions (lines 236-238).

**3. Methodology.**
**R2:** *Thanks to the restructuring performed by the authors, this part is now much easier to read and understand. To further improve it, I suggest clarifying the following:*

*It is not so immediately clear to the reader if the authors built their impact chain from zero following the methodology developed in the Paratus Project, or if they used as a starting point an Impact Chain developed in the Paratus project but based on the methodology developed outside the project by Eurac. I mainly refer to this unclear sentence: "Building the Impact Chain initially developed within the Paratus Project" (lines 201-202). I invite the authors to clarify this aspect from the beginning.*

**Response:** The authors of the paper are the same as the ones of the initial Impact Chain, that was built within the Paratus project. The enhanced version of this Impact Chain is a continuation of our work that stepped outside the project. We thank the reviewer for highlighting this misunderstanding that was amended at lines 292-294:
*The next section presents two distinct workflows within the methodological framework (Figure 3): building the Impact Chain initially developed by the authors within the Paratus Project (PARATUS Deliverable 1.1 2023) – which was further strengthened by first responders' input and, secondly, its enhancement to account for vulnerability augmentation.*

**R2:** *The synthetic explanation of the Impact Chain construction in Section 3.1 lacks mention of all the steps depicted in Fig. 3. Furthermore, even when these steps are referenced, the text doesn't consistently use the labels that appear in the figure. To enhance the clarity of the methodology, I encourage the authors to ensure better alignment between the text in Section 3.1 and the content depicted in panel 1 of Figure 3.*

**Response:** We express our gratitude to the reviewer for this meticulous and important observation. We revised both Section 3.1 and Figure 3 and provided an updated description of the methodology related to the building of the Impact Chain (lines 315-373).

**R2:** *Section 3.2 would benefit from minor restructuring to enhance reader comprehension of the steps outlined. The authors wrote "This broadening of the original application of the Impact Chain was done by 1) introducing new types of elements (i.e., augmented vulnerabilities, derived impacts), 2) establishing new types of connections between the impacts/adaptation options and vulnerabilities, and 3) ranking the vulnerabilities in the Impact Chain based on their augmentation. These steps were implemented to construct an enhanced Impact Chain, building on the*

*previous version that documented the unfolding of the selected co-occurrent hazards in Romania in 2020-2021.", but then, a few lines later, they start saying that "The first step was to perform an in-depth analysis of the vulnerabilities in the Impact Chain.". To ensure coherence, it is essential to present the steps in the correct order, and, also in this case, to refer more to the steps reported in Fig.3. Additionally, it would be helpful to introduce and define the new types of elements (i.e., augmented vulnerabilities, derived impacts) earlier in the text, perhaps before discussing the new types of connections. This anticipatory approach can enhance clarity and understanding for the reader.*

**Response:** The Methodology section was revised according to the guidelines. We listed all the steps in the right order (lines 381-445). However, to introduce the meaning of augmented vulnerabilities or derived impacts earlier in the paper or section (although having the merits mentioned by the reviewer) would contradict the order of the steps in Figure 3. First, we have to present the connections that express the augmentation, and then, based on those connections, we can identify the augmented vulnerabilities and the derived impacts. Introducing these elements earlier would be confusing to the reader without firstly presenting the connections. Nevertheless, we hope that Section 3.2 reads better now and that the augmentation framework is easier to understand. Please note that we also revised the explanation of derived impacts (lines 418-428) to facilitate its understanding.

**4. Results**
**R2:** *Lines 398-401: "Some vulnerabilities underwent multiple transformations into derived impacts […] The explanation lies in the fact that multiple impacts can augment the same vulnerability, creating also a derived impact that reinforce the impact that generated the augmentation." The concept expressed in these sentences is not so easily understandable. I invite the authors to consider rephrasing it.*

**Response:** We appreciate this insightful observation which was also brought forward by the other reviewer. We amended the explanations about derived impacts throughout the paper (lines 418-428, 522-535). The new paragraph highlighted by the reviewer now reads: *When augmented, certain vulnerabilities can function similarly to impacts, reinforcing the very impact that initially increased the vulnerability, forming a positive loop feedback composed of "deepens/shifts" links and "sharpens" links. Such augmented vulnerabilities with double status were duplicated in the enhanced Impact Chain and labelled as "derived impacts", as detailed in Appendix B. Some vulnerabilities underwent multiple transformations into derived impacts because they acted as (derived impacts) in relation to more than one augmentation-generator impact (Figure 6). This resulted in a larger number of cases where the augmentation of a vulnerability created a derived impacts (15 cases) compared to the number of actual derived impacts (9) in the chain.*

**R2:** *Despite "the intricate configuration of the Impact Chain does not allow for a proper visualisation within this paper", I think that it would be highly beneficial, even essential, to include some graphical representations of the derived impacts described verbally in Section 4.3. This section constitutes the core of the work and would be communicated more clearly and efectively by introducing some 'extracts' from the Impact Chains to aid in visualization.*

**Response:** This is a valuable suggestion, and we are thankful for it. Indeed, Section 4.3 Derived impacts would benefit from visual support. This was included as the new Figure 6, and the text of the section is now linked to different parts of this figure, facilitating the understanding of the mechanism of derived impacts.

**R2:** *Moreover, I strongly suggest the authors double-check grammar and typos.*

**Response:** We thank the reviewer for flagging this issue. Please note that the paper was spell-checked and reread again, and the identified errors were corrected, as can be seen throughout the text.

Respectfully yours,
The Authors